# Investigating the Long-term Variability of the Red Sea Marine Heatwaves and their Relationship to Different Climate Modes: Focus on 2010 Events in the Northern Basin

**Manal Hamdeno[1,2], Aida Alvera-Azcárate[1], George Krokos[3,4], Ibrahim Hoteit[3]**

[1]GeoHydrodynamics and Environment Research (GHER), University of Liège, Liège, Belgium
[2]Oceanography Department, Faculty of Science, Alexandria University, Alexandria, Egypt
[3]Physical Sciences and Engineering Division, King Abdullah University of Science and Technology (KAUST), Thuwal, Saudi Arabia
[4]Institute of Oceanography, Hellenic Centre for Marine Research, Anavyssos, Greece,

*Correspondence to*: Manal Hamdeno (mh.elawady@uliege.be)

**Abstract.**

Marine heatwaves (MHWs) have been increasing in frequency and intensity worldwide and posing a serious threat to marine ecosystems and fisheries. The Red Sea (RS), a semi-enclosed marginal sea, is highly vulnerable to climate change due to its small volume and slow rate of water renewal. Despite the importance of the RS, MHWs in this region remain poorly studied and understanding of their spatial and temporal characteristics and forcing mechanisms is limited. This study examines MHWs in the RS over the last four decades (1982-2021) and their relationship to large-scale climate modes, with particular focus on the 2010 MHW event in the northern Red Sea (NRS). Analysis of sea surface temperature anomaly (SSTA) trends in the RS revealed a decadal variability, with the highest warming trends occurring alternately in the northern and southern regions. The RS has experienced a significant warming trend over the last four decades, which has intensified after 2016. This warming has led to an increase in the frequency and duration of MHWs in the region, with 46% of events and 58% of MHW days occurring only in the last decade. The RS exhibits a meridional gradient, with decreasing mean annual MHW intensity and duration, but increasing mean annual MHW frequency from north to south. The annual MHW frequency in the NRS peaked in 2010, 2018, 2019 and 2021, while in the Southern Red Sea (SRS) the highest frequency occurred in 1998 and from 2017 to 2021. The study also examined the correlation between MHWs and climate indices and found that the Atlantic Multidecadal Oscillation (AMO), the Indian Ocean Dipole (IOD) and the East Atlantic/Western Russia pattern (EATL/WRUS) were the three dominant modes that correlated with SSTA and MHWs in the region. The North Atlantic Oscillation (NAO) and the Oceanic Niño Index (ONI) showed weaker and less significant correlations. Finally, the authors conducted a case study of the 2010 MHW event in the NRS, which was the most intense and longest winter event of the year. Using a high-resolution ocean model and atmospheric reanalysis data, it was found that the late winter 2010 MHW in the NRS extended to a depth of 120 meters and was driven by a combination of atmospheric forcings, particularly an increase in air temperature (Tair) and humidity, possibly linked to reduced winds leading to reduced latent heat flux (LHF) and strong ocean warming, creating favorable conditions for MHW to occur.

## 1 Introduction

Episodes of very warm sea surface temperature (SST), known as "marine heatwaves" (MHWs), have been observed in all the world's oceans and marginal seas and have increased in frequency, intensity and duration in the recent decades (Hobday et al., 2016, 2018; Oliver et al., 2018, 2021; Holbrook et al., 2020; Sen Gupta et al., 2020). These extreme warm water events can be triggered by atmospheric forcings, oceanic processes, large climate variability, or a combination of them, and these drivers can vary based on season and geographic location (Holbrook et al., 2019; Amaya et al., 2020; Sen Gupta et al., 2020; Mohamed

et al., 2021, 2022, 2023; Oliver et al., 2021; Pujol et al., 2022). MHWs have ecological and socioeconomic impacts, including coral bleaching (Eakin et al., 2010; Hughes et al., 2018; Genevier et al., 2019), declines in sea surface productivity (Le Grix et al., 2021; Hamdeno et al., 2022; Hamdeno and Alvera-Azcaráte, 2023), mortality of benthic communities (Garrabou et al., 2009; Smale and Wernberg, 2009; Rivetti et al., 2014), and loss of seagrass beds (Diaz-Almela et al., 2007; Carlson et al., 2018; Chefaoui et al., 2018) and kelp forests (Arafeh-Dalmau et al., 2019).

The increasing risk of MHWs to ecosystems and economies requires a thorough understanding of their causes, especially in vulnerable areas such as the Red Sea (RS), which harbors a unique ecosystem and is of great political and economic importance (Meziere et al., 2021). The RS is a semi-enclosed, elongated marginal sea between Africa and Asia, connected to the Indian Ocean on the south by the Strait of Bab-al-Mandeb and to the Mediterranean Sea on the north by the Suez Canal (Fig. 1). Due to their small volume and slow rate of water renewal, marginal, semi-enclosed seas such as the RS are particularly vulnerable

to global warming (Belkin, 2009). The RS has an arid climate and a negative water balance, i.e. evaporation exceeds precipitation and river runoff combined (Bower and Farrar, 2015; Eladawy et al., 2017; Liu and Yao, 2022). The RS is a vital resource for fisheries, agriculture, tourism, and freshwater production through desalination, and is a major shipping route (Barale, 2014; Hoteit et al., 2021).

Due to the importance of the RS, several studies have investigated its physical properties, especially SST, as it can serve as an

indicator of the thermal stress caused by global warming, which has devastating effects on the rich and diverse marine life of the RS (Raitsos et al., 2011; Barros et al., 2014; Chaidez et al., 2017; Karnauskas and Jones, 2018; Shaltout, 2019; Trisos et al., 2020; Liu and Yao, 2022). However, only a few studies have investigated MHWs in the RS (Genevier et al., 2019; Bawadekji et al., 2021; Mohamed et al., 2021), and up to date there have been no studies investigating the link between climate patterns and the occurrence of MHWs in the RS region. Raitsos et al. (2011) used satellite-derived SST to examine the

spatiotemporal changes in RS temperatures between 1985 and 2007 and concluded that the RS is experiencing a strong warming that began in the mid-1990s and increased abruptly after 1994. Another study by Chaidez et al. (2017) calculated the warming trends in the RS between 1982 and 2015 and estimated the overall rate of warming for the RS to be 0.17 ± 0.07 °C/decade, while the SST trend in the NRS was between 0.40 and 0.45 °C/decade, which is higher than the global rate. A recent study by Liu and Yao (2022) examined the long-term variations in the SST of the RS and adjacent seas, as well as

surface air temperatures, from 1875 to 2019. They found that the SST of the RS increased at an average rate of 0.43 °C/decade during these years, and this rate has accelerated in recent decades. Furthermore, they found that the SST anomalies of the RS

are positively correlated and strongly synchronized with those of the adjacent seas and with air temperature anomalies. According to the IPCC, the RS is warming due to climate change and a temperature rise of 3.45 °C is expected for the period 2010-2099 (Barros et al., 2014; Trisos et al., 2020).

Genevier et al. (2019) established a link between the MHWs and coral bleaching in the RS during the summer months (July-October). Using satellite SST data from 1985 to 2015, they found that MHWs in the RS can trigger coral bleaching when their SSTs exceed the 95[th] percentile, when the climatological average is 30°C or higher, last for at least seven consecutive days and occur at shallow depths (< 150 m). Bawadekji et al. (2021) studied the general and local characteristics of the MHWs in the RS. Their study concluded that the RS exhibits a meridional gradient with decreasing average annual MHW intensity and

MHW duration from north to south, and a meridional gradient with increasing average annual MHW frequency from north to south. Finally, Mohamed et al. (2021) studied the spatio-temporal variability and trends of MHWs in the RS over 39 years (1982-2020) using high-resolution satellite SST data. Their results showed that over the last two decades (2000 – 2020), the average frequency and duration of heatwaves increased by 35% and 67%, respectively. Their study also showed that the highest annual MHW frequencies were recorded in 2010, 2017, 2018, and 2019.

It is well known that the oceans are the main drivers of internal climate variability, affecting climate around the world (e. g., Shukla, 1998). Many modes of climate variability are coupled ocean-atmosphere phenomena, such as the El Nino-Southern Oscillation (ENSO), which is the most important mode of global climate variability on interannual time scales (e.g., McPhaden et al., 2006). At the local scale, it has long been known that the broader region surrounding the RS is directly influenced by the North Atlantic Oscillation (NAO) (Visbeck et al., 2001), and the Atlantic Multidecadal Oscillation (AMO) (Krokos et al.,

2019), while ENSO is known to indirectly influence the neighboring tropical Indian Ocean via atmospheric teleconnections (Bjerknes, 1969).

The AMO was identified as a coherent mode of natural variability in the North Atlantic with an estimated duration of 60-80 years (Zhang, 2007; Semenov et al., 2010). It is based on average SST anomalies in the North Atlantic basin, typically over 0° – 80° N (Semenov et al., 2010; Schneider et al., 2013). The positive AMO phase corresponds with positive SST anomalies

over most of the North Atlantic, with stronger anomalies in the subpolar region and weaker anomalies in the tropics. It has significant regional and hemispheric impacts on climate, such as the Northern Hemisphere mean surface temperature (Zhang, 2007; Semenov et al., 2010; Schneider et al., 2013). The Indian Ocean Dipole (IOD) is a climate phenomenon that occurs in the Indian Ocean, and is defined as the difference in SST between the eastern and western regions of the Indian Ocean. The IOD can significantly affect weather patterns and climate in surrounding regions, including parts of Africa, Southeast Asia,

and Australia. During a positive phase, warm water is pushed into the western part of the Indian Ocean, while cold deep water rises to the surface in the eastern Indian Ocean and vice versa during the negative phase (Behera et al., 2021; Cai et al., 2021). The East Atlantic/ West Russia (EATL/WRUS) pattern is one of three prominent teleconnection patterns that affect Eurasia throughout the year. The major surface temperature anomalies associated with the positive phase of the EATL/WRUS pattern reflect above-average temperatures over East Asia and below-average temperatures over much of western Russia and

northeastern Africa (Barnston and Livezey, 1987). The NAO index is based on the sea level pressure difference between the

subtropical high (Azores) and the subpolar low. Strong positive phases of the NAO are usually accompanied by above-average temperatures in the eastern United States and northern Europe and below-average temperatures in Greenland and often in southern Europe and the Middle East (Barnston and Livezey, 1987; Dool et al., 2000; Chen and Dool, 2003). ONI is a primary indicator for monitoring El Niño and La Niña, which are opposite phases of the climate pattern called El Niño-Southern Oscillation "ENSO". The Oceanic Niño Index (ONI) is the difference between a three-month running average of SST over an ocean area between 120° W and 170° W along the equator and the long-term average for the same three months. El Niño conditions are considered to exist when the Oceanic Niño Index is +0.5 or higher, meaning that the eastern and central tropical Pacific Ocean is significantly warmer than normal. La Niña conditions are present when the Oceanic Niño Index is -0.5 or lower, meaning that the region is cooler than normal (Bamston et al., 1997; Hoerling et al., 2001; McPhaden et al., 2006; Huang et al., 2017).

The objective of this study is to conduct a comprehensive analysis of the spatial and temporal variability of MHWs in the RS and to identify its regional patterns. The study also aims to investigate the potential links between various climate modes, particularly the AMO, the IOD, the EATL/WRUS pattern, the NAO and the ONI, with the annual sea surface temperature anomaly (SSTA) and the annual frequency of MHWs in the RS. As a case study, we will focus on the 2010 MHWs and provide a detailed description of the spatial and vertical extent and potential atmospheric drivers of the intense event in that year. The motivation for selecting 2010 as a case study is that it was one of the warmest years with highly frequent MHWs and had a different spatial distribution of SSTA and marine heatwave days (MHWDs) than the other warm years.

To accomplish these goals, the study is structured into four main sections. The first section focuses on the characteristics and trends of SSTs and MHWs in the RS between 1982 and 2021. The second section examines the interannual variability of SSTA and MHWs over the last four decades in the RS and its northern and southern basins. The third section explores the relationship between SSTA/MHWs in the RS and the different climate modes. Finally, in the fourth section, a case study of the 2010 MHW events in the NRS is presented.

## 2 Data and Methods of Analysis

### 2.1 Datasets

To analyze the spatial and temporal variability of SST and MHWs in the RS and examine their interactions with different climate modes, focusing on the 2010 MHW events as a case study, various available data sources are used:

i-  RS bathymetry was obtained from GEBCO's current bathymetric dataset, the GEBCO_2023 Grid (https://www.gebco.net/data_and_products/gridded_bathymetry_data/). This is a global terrain model for ocean and land that provides elevation data in meters on a grid with an interval of 15 arc-seconds (Schenke, 2013). The bathymetry of the RS was extracted from the global bathymetry map.

ii-  Daily high-resolution SST data from January 1, 1982 to December 31, 2021 obtained from the Copernicus Marine Environment Monitoring Service (CMEMS; https://data.marine.copernicus.eu/product/SST_GLO_SST_L4_REP_OBSERVATIONS_010_011/description)

website. The CMEMS Operational SST and Ice Analysis (OSTIA) reprocessed analysis product is based on an SST satellite and in situ observation (Good et al., 2020). The SST dataset consists of daily, gapless maps of SST and ice concentration (referred to as the L4 product) with a horizontal grid resolution of $0.05° \times 0.05°$.

iii- Hourly mixed layer depth (MLD) and water column temperature are obtained from a regionally tuned simulation of the MIT general circulation model (MITgcm; Marshall et al., 1997) with a horizontal resolution of 1 km and 50 vertical layers (Krokos et al., 2021). The model domain covers the entire RS, including the two Gulfs (Suez and Aqaba) at the northern end, with an open boundary in the Gulf of Aden. The topography of the model is based on the General Bathymetric Map of the Ocean (Weatherall et al., 2015) updated with available regional data. The model is driven with hourly, high-resolution (~5km) atmospheric downscaled WRF fields (Viswanadhapalli et al., 2017). The results of the MITgcm model for the RS have been extensively validated against different data sets and in different environments and applications, as described in Hoteit et al., (2021) and Krokos et al., (2021).

iv- The normalized monthly oceanic El Niño-Southern Oscillation Index (ONI), East Atlantic/West Russian Pattern (EATL/WRUS), Atlantic Multidecadal Oscillation (AMO), and North Atlantic Oscillation (NAO) time series from 1982 to 2021 were obtained from the National Oceanic and Atmospheric Administration (NOAA) (https://psl.noaa.gov/data/climateindices/list/). The Indian Ocean Dipole (IOD) was downloaded from the Japan Agency for Marine-Earth Science and Technology (JAMSTEC) for the aforementioned period (https://www.jamstec.go.jp/virtualearth/general/en/).

v- Hourly atmospheric data is used to examine the variability of atmospheric conditions in relation to the variability of the SSTA over the entire study period, and used to examine the drivers of the 2010 MHW events, are from the European Center for Medium-Range Weather Forecasts (ECMWF) ERA5 ((Hersbach et al., 2020); https://cds.climate.copernicus.eu/cdsapp#!/dataset/reanalysis-era5-single-levels). The dataset has a spatial resolution of $0.25° \times 0.25°$. The atmospheric fields include the wind components at 10 m altitude (U10 and V10), air temperature at 2 m altitude (Tair), dew point temperature at 2 m altitude (d2m), mean sea level pressure (MSLP), shortwave surface net radiation ($Q_s$), longwave surface net radiation ($Q_b$), sensible surface heat flux ($Q_h$), and latent surface heat flux ($Q_e$). Daily mean values of atmospheric variables were calculated by averaging the hourly data.

## 2.2 Methods of Analysis

MHWs can be characterized by different methods, each of them has its advantages and disadvantages. These methods include the use of fixed thresholds, relative thresholds, and seasonally varying thresholds (Hobday et al., 2016; Mohamed et al., 2022). In this work, the approach of Hobday et al. (2016, 2018) was used to define and categorize the RS surface MHWs (between 1982 and 2021) and subsurface MHWs (for the February-March 2010 MHW event in the northern region as a case study). Hobday et al. (2016) defined a MHW as an event of unusually high water temperature lasting five consecutive days or longer. During the MHW, water temperature exceeds the 90th percentile climatological threshold. The climatological mean and threshold are calculated in each grid cell for each calendar day of the year using daily water temperature data (at the surface and subsurface levels of the water column) over a 40-year period (1982-2021). MHWs can be described with a number of metrics, which are their duration (in day), which refers to the period between the start and end dates of a MHW event, frequency (in events) indicates the number of MHW events that have occurred within a given year or period, mean intensity (°C) is the average value of the temperature anomaly during the duration of a MHW event, maximum intensity (°C) is the highest value of the temperature anomaly recorded during a MHW event, cumulative intensity (°C.day) is the integrated temperature

anomaly over the entire duration of a MHW event and is a measure of the overall intensity of the event, and total MHW days (MHWDs, in day) refers to the total number of MHW days that have occurred in a given year/period (Hobday et al., 2016, 2018). The MATLAB toolbox M_MHW was used to define the MHW metrics (Zhao and Marin, 2019). To provide a more comprehensive and detailed description of MHWs in the RS, we have divided the RS into two regions: the Northern Red Sea

(NRS) and the Southern Red Sea (SRS). The NRS extends from 22°N to 30°N, while the SRS extends from 22°N to 12.5°N. This division was based on the north-south spatial thermal gradient in the RS, which shows different characteristics of SST and MHWs between the two sub-basins.

The winter and summer SST in the RS was calculated and averaged over the study period (1982-2021) at each grid point. The winter season was represented by the months of January, February, and March, while the summer season was represented by

the months of July, August, and September. The selection of winter and summer months was based on the seasonal cycle of SST, with the three months of the lowest SSTs representing the winter season and the three months of the highest SSTs representing the summer season. We focused on these two seasons as it was observed that the most intense RS MHWs occurred predominantly during winters and summers. SST anomalies were calculated by removing the historical climatological mean (1982-2021) at each grid point from the SST values at the same location. The strong seasonal signal was removed from the

SSTA data at each grid cell to obtain a deseasonalized map and time series (Skliris et al., 2012). Linear trends in SSTA and MHW metrics are estimated using the least squares method (Wilks, 2019) and their statistical significance is determined using the Modified Mann-Kendall test (MMK) at the 95% confidence level, which takes autocorrelation into account when assessing the significance of the trend (Hamed and Ramachandra Rao, 1998; Wang et al., 2020).

We further investigated the characteristics of MHWs during 'warm' or 'cold' periods. Specifically, we define warm periods as

those that exhibit a pronounced positive SSTA compared to the long-term average, while cold periods are characterized by a pronounced negative SSTA. Warm years are identified as those that are warmer than the preceding and following year, and cold years as those that are colder than the year before and after. The definition of "cold" and "warm" years is relative to the SSTA variability and does not necessarily imply that the SSTA in those years was unusual or extreme. Once the warm or cold years are identified, we calculate the average SSTA for those years by averaging the SSTA values over the entire years for

each grid cell. Similarly, we calculate the MHWDs for the warm or cold years by averaging the MHWDs over those years for each grid cell. This gives us an indication of the overall spatial variability of the SSTA/MHWDs during the warm or cold years in our study period.

To gain a deeper understanding of the relationship between the different climate modes and the occurrence of MHWs over the last four decades in the RS, spatial correlations were examined. The climate modes considered in this study are the Oceanic

Niño Index (ONI), the East Atlantic/West Russia Pattern (EATL/WRUS), the Atlantic Multidecadal Oscillation (AMO), the North Atlantic Oscillation (NAO) and the Indian Ocean Dipole (IOD). The correlation maps were calculated using the Pearson correlation coefficient (r), a widely used method for measuring linear correlations between two variables (Kirch, 2008; Patten and Newhart, 2017). The Pearson correlation coefficient ranges from -1 to 1, where -1 stands for a perfect negative correlation, 1 indicates a perfect positive correlation and 0 for no correlation. In this study, we calculated the correlation maps between the

annual time series of each climate mode and the annual MHWs/SSTA in the RS. The MHWs were identified and characterized using a set of metrics, such as their duration, frequency, mean intensity, maximum intensity, cumulative intensity and total days (as described in the previous Section). We calculated the correlation between the different annual climate modes and annual MHW metrics, in particular frequency, duration and total days. The results showed a consistent spatial correlation pattern with MHW different metrics, while the correlation coefficients varied only slightly. For the sake of brevity, we present

only the correlation results with MHW frequency and SSTA in our results. The MHW frequency was chosen for presentation due to its slightly higher correlation compared to MHW duration and total days. To test the significance of the correlations, a two-tailed t-test was used (Patten and Newhart, 2017). The t-test is a statistical hypothesis test that compares the means of two samples and determines whether they differ significantly from each other. Finally, we also compared the time series between different climate modes and the frequency of MHWs in the RS and its sub-basins. By analyzing the correlation maps and the

significance of the correlations, we can gain insights into the potential co-variability between MHWs in the RS and larger-scale climate variability.

In this study, we have selected MHW events that occurred in 2010 for detailed analysis, focusing particularly on the most intense and longest winter event occurred in that year. The MHW events were identified using the methodology described by Hobday et al., (2016) and then categorized into four intensity levels based on the multiple of the local difference between the

climatological mean and the climatological 90$^{th}$ percentile, which serves as a threshold for identifying MHW. This anomaly varies by location and time of year. The magnitude of the scale descriptors was defined as follows: moderate (1–2×, category I), strong (2–3×, category II), severe (3–4×, category III) and extreme (>4×, category IV) (Hobday et al. 2018). The intense event (category three MHW) that occurred between February and March 2010 was studied in detail, including its vertical extent and potential atmospheric drivers. The MHWs at different levels of the water column (0m, 25m, 55m, 110m and 130m

depth) were calculated using daily averages of hourly modeled water temperature data. The vertical extent of MHW was then defined by the first depth without MHWs. We also investigated the atmospheric conditions associated with these MHW events by using the ERA-5 atmospheric data to better understand the potential triggers for this event. Following the work of Thomson and Emery (2014) and the description of Nagy et al. (2017, 2021), the net surface heat flux ($Q_T$) in W/m$^2$ was calculated as expressed in Eq.1:

$$Q_T = Q_s + Q_b + Q_c + Q_e \,, \tag{1}$$

where $Q_s$ is the heat absorbed by the ocean from incident solar radiation in W/m$^2$, $Q_b$ is the heat loss from back radiation in W/m$^2$, $Q_c$ is the sensible heat loss from convection and conduction in W/m$^2$, and $Q_e$ is the heat loss from evaporation (latent heat) at the ocean surface in W/m$^2$.

According to Alduchov and Eskridge, (1996) and as described in Bawadekji et al., (2022), the relative humidity (RH) were

calculated from ERA5 air temperature (Tair) and dew point temperature (d2m) as expressed in Eq.2:

$$RH = \frac{100*\exp\left(\frac{17.625*d2m(°C)}{243.04+d2m(°C)}\right)}{\exp\left(\frac{17.625*Tair(°C)}{243.04+Tair(°C)}\right)} \,, \tag{2}$$

## 3. Results and Discussion

### 3.1 SST and MHWs characteristics and trends in the Red Sea (1982 – 2021)

The spatial maps of average SST over the entire study period (1982-2021), winter months (January, February and March) and summer months (July, August and September) in the RS are shown in Figure 2. The average SST in the RS was between 23 and 28 °C throughout the study period, with a meridional gradient from north to south, with the highest temperatures observed in the SRS and the lowest in the NRS and the Gulfs of Suez and Aqaba (Fig. 2a). In the winter (JFM), the average SST fluctuated between 18 and 27 °C (Fig. 2b), while in the summer (JAS) it fluctuated between 26 and 32 °C (Fig. 2c).

The marine environment is influenced by both natural variability and global warming trends. Over time, the difference between a fixed baseline and current temperatures can widen, leading to an increasing number of detected MHWs. This temporal shift can complicate long-term studies of MHW trends and their impacts on marine ecosystems. However, using a fixed baseline simplifies the methodology for detecting MHWs and avoids the complexity that could arise from periodically updating the baseline, which could introduce variability and reduce the clarity of the detection process. A fixed baseline provides a standardized reference period that ensures consistency of MHW detection and analysis across different studies and time periods (e.g. (Genevier et al., 2019; Cheng et al., 2023)). This consistency allows for straightforward comparison between MHW events detected using the same criteria. In addition, a fixed baseline serves as a historical benchmark against which current and future MHW events can be measured. This allows an assessment of how current conditions deviate from a known historical norm. To account for SST variability during the study period and to emphasize the impact of external forcing on marine ecosystems, we calculated MHW characteristics in the RS based on 40-years of climatology (1982-2021).

The RS exhibits a high spatial variability of MHW characteristics. The mean annual MHW frequency varied between 1.5 and 2.5 events, with the highest mean frequency values recorded in the coastal areas of the SRS and the Strait of Bab El-Mandab (Fig. 3a). The mean duration of the MHW ranged from 8 to more than 20 days (Fig. 3b), with longer MHW durations observed in the NRS and the Gulfs of Suez and Aqaba. The mean and maximum annual MHW intensities ($I_{mean}$ and $I_{max}$) showed the same pattern of spatial distribution with slightly different magnitudes (Fig. 3c, d). The most intense MHWs were observed in the NRS and in the western part of the SRS around Dahlak Kebir Island. Furthermore, the mean cumulative MHW intensity ($I_{cum}$) and the total number of heat days exhibited a north-south gradient, with higher values in the NRS region (Fig. 3e, f). Figure 3e shows that the mean MHW cumulative intensity varied between 10 and 35 °C.days, with the highest values (> 30 °C. days) found in the NRS and the lowest in the SRS and the Strait of Bab El-Mandab. The mean total MHW days ranged from 20 to over 30 days, with the highest values found in the NRS, coastal areas of the SRS and the Gulfs of Suez and Aqaba. (Fig. 3f). In general, the MHWs in the RS showed different characteristics between the northern and southern basins. The MHWs of the NRS were longer and more intense than those of the SRS, while the MHWs of the SRS were characterized by their frequent occurrence. The same pattern of MHW distribution was also observed by (Bawadekji et al., 2021 and Mohamed et al., 2021).

Spatial trend maps of deseasonalized SST and MHW frequency from 1982 to 2021 in RS are shown in Figure 4. A statistically significant ($p < 0.05$) trend with a 95% confidence interval was observed across the region. The trends of SST and MHW frequency in the RS are not uniform and range between 0.1 to 0.5°C/decade for SST and 0.5 to 2 events/decade for MHW frequency as shown in Figure 4. The NRS experienced high SST trends, with a maximum of about 0.4°C/decade between 25°N and 28°N. However, the highest SST trends were observed between 16°N and 25°N, with a gradient that increased towards offshore waters and exceeded 0.45°C/decade. The Strait of Bab El-Mandab and the Gulfs of Suez and Aqaba, on the other hand, showed the lowest SST trends, with trends below 0.15°C/decade. The MHW frequency trends were found to be highest in the SRS and the Gulfs of Suez and Aqaba, with a rate of over 1.5 events/decade, while the lowest trends were observed in the NRS and the Strait of Bab El-Mandab, with a rate of less than 1 event/decade. Furthermore, the trends in MHW duration and total days also displayed spatial variation in the RS (Supplementary Figure S1). The MHW duration spatial trend fluctuated between 2 to more than 10 days per decade, with an average temporal trend of $2.8 \pm 1.25$ days/decade (Fig. S1 a-b). The highest MHW duration trends were observed in the central RS, some parts of the NRS, and the northern part of the Suez and Aqaba Gulfs. The MHW total days trend ranged between 15 to more than 30 days per decade, with an average temporal trend of $20.04 \pm 6.88$ days/decade (Supplementary Figure S1 c-d). The highest trends were observed in the SRS, some parts of the NRS, and the northern part of the Suez and Aqaba Gulfs. Notably, for all the MHW metrics (frequency, duration, and total days), there is an accelerated positive trend that is more pronounced in the last decade.

**3.2 SST and MHWs interannual variability**

The temporal SSTA trends over the study period were $0.33 \pm 0.02$ °C/decade, $0.34 \pm 0.04$ °C/decade and $0.32 \pm 0.03$ °C/decade for the entire RS, NRS and SRS, respectively (Fig. 5). Our results are in good agreement with (Raitsos et al., 2011; Barros et al., 2014; Chaidez et al., 2017; Liu and Yao, 2022). The analysis of SSTA between 1982 and 2021 revealed three distinct phases of variability in the RS and its sub-basins (Fig. 5). The first phase, from 1982 to 1992, was characterized by negative SSTA on average. The second phase, between 1993 and 2015, showed a slow warming trend, but the SSTA fluctuated around zero, suggesting a relatively stable period with increased inter-annual variability. The third phase, from 2016 to 2021, was marked by a rapid increase in SSTA, with the anomaly consistently remaining positive. Moreover, the monthly SSTA data for the RS and its sub-basins show a clear warming trend that began in 1994, with an initial SSTA of approximately 0.5°C. The SSTA remained relatively stable for a few years, but then increased rapidly after 2016, reaching an SSTA of 1.5°C or higher (Supplementary Figure S2.a-c). This finding is consistent with the results of Raitsos et al. (2011). It was also observed that during years when the RS experienced cold SSTs, the NRS was warmer than the SRS, especially during the winter and autumn months (Supplementary Figures S2. d). Conversely, during years when the RS experienced warm SSTs, the SRS was warmer than the NRS (Supplementary Figures S2. d). The year 2010 was one of the warmest years in the RS, but it was particularly warm in the NRS, with an SSTA difference between the NRS and SRS of over 1°C (Supplementary Figures S2. d).

Further analysis revealed a decadal variability in the SSTA trends (Supplementary Figures S3-4). Between 1982 and 1991, the highest trends were observed in the NRS, with an average trend of 0.56°C/Decade, while the average trend in the SRS was

0.26°C/Decade. From 1992 to 2001, the spatial pattern of the SSTA trend was altered, with the highest trends observed in the SRS, with an average trend of 0.57°C/Decade, and lower trends in the NRS, with an average trend of 0.30°C/Decade. From 2002 to 2011, the highest trends were again observed in the NRS, with an average trend of 0.45°C/Decade, while the SRS

experienced no trend in the SSTA during this period. Finally, over the last decade of the study period (2012-2021), the SRS had higher trends in the SSTA than the NRS, with an average trend of 1.35°C/Decade for the SRS and 0.89°C/Decade for the NRS.

Over the study period, there were years that were notably colder or warmer than the average for that period. In this study, we defined "cold years" as those that were colder than both the previous and following year, and "warm years" as those that were

315 warmer than the previous and following year. The cold years were 1985, 1990, 1992, 1993, 1997, 2012 and 2013, while the warm years were 1991, 1995, 2010 and the last six years of the study period (2016-2021). During both the warm and cold years, the spatial distribution of the average SSTA and MHWDs was analyzed, as shown in Figures 6-8. In the cold years, the NRS and the Strait of Bab El-Mandab had the highest SSTA and MHWDs (Figure 6). However, in the warm years, the SRS had the highest SSTA, and the SRS and the northern regions of the Gulfs of Suez and Aqaba had the highest number of

320 MHWDs (Figure 7). The year 2010 was an exception among the warm years, with a distinct spatial distribution of SSTA and MHWDs. In 2010, the NRS and the Gulfs of Suez and Aqaba had the highest SSTA and MHWDs (Figure 8).

The inter-annual variations of MHWs frequency in the RS and its sub-basins during the study period are shown in Figure 9. The threshold for determining the years with the highest MHW frequency was set at one standard deviation above the mean of the annual MHW frequency. Based on this criterion, any year with more than four MHW events was classified as having a

325 high MHW frequency. In the entire RS, the year 2010 and the last five years of the study period (2017-2021) had the highest annual MHW frequency (Figure 9a). The NRS experienced the highest annual MHW frequency in the years 2010, 2018, 2019, and 2021 (Figure 9b), while in the SRS, the year 1998 and the last five years of the study period had the highest annual MHW frequency (Figure 9c). A total of 78 MHW events were recorded in the RS over the last four decades (1982-2021), with 36 of these events (46%) occurring in the last 10 years of the study period. Furthermore, a total of 1016 heat days were observed in

the RS between 1982 and 2021, with 590 of these days (58%) occurring in the last decade. The findings of this study suggest that the recent rapid increase in SST in the RS has led to a positive trend in the occurrence of MHWs in the region. These findings are consistent with those of previous studies, such as Bawadekji et al., (2021) and Mohamed et al., (2021), which have also documented the increasing trend of MHWs in the RS. This trend is expected to continue in the future, as global warming is projected to cause further increases in SSTs, both in the RS and in other regions around the world.

To gain a better understanding of the atmospheric conditions associated with the years with the highest frequency of MHWs in the RS and its subregions, we compared the atmospheric variables with the annual MHW frequency. The annual anomalies of Qt, Tair and wind speed (Ws) are shown in Supplementary Figures S5-S7. Our analysis revealed that the years with a high MHW frequency were characterized by a specific set of atmospheric conditions. In particular, these years were characterized by reduced Ws and high Tair and Qt which may have contributed to the frequent occurrence of MHWs in the RS during these

340 years.

### 3.3 Climate modes and MHWs in the Red Sea

To investigate the potential relationship between climate indices, annual sea SSTA, and MHW metrics (i.e., frequency, duration, and total days) in the RS over the past four decades, a correlation analysis was conducted between different climate modes and SSTA/MHW metrics. It was observed that the correlation between climate modes and MHW frequency, duration, and total days displayed similar spatial patterns with slightly different correlation values that were not significantly different. To avoid repetition of figures and text, this section presents the correlation of climate modes with MHW frequency only. The significance of the correlations was tested with a 95% confidence interval. The analysis focused on the NOAA modes that showed a significant correlation ($p < 0.05$) with the SSTA and/or MHW frequency, as presented in Figure 10.

The AMO index showed a highly significant positive correlation with both SSTA and MHW frequency across the whole RS, with a correlation coefficient of greater than 0.7 for SSTA and greater than 0.5 for MHW frequency (Figures 10a and b). This finding is consistent with a previous study by Krokos et al. (2019), who reported that long-term AMO oscillations modulate SST trends in RS. The IOD index also showed a positive correlation with both SSTA and MHW frequency in RS, with a correlation coefficient of greater than 0.5 for SSTA and ranging between 0.2 and 0.4 for MHW frequency (Figures 10c and d). The influence of the IOD on SSTA and MHW frequency was stronger in the SRS than in the NRS. A strong negative correlation was observed between the EATL/WRUS index and both SSTA and MHW frequency in the RS, with a correlation coefficient of approximately -0.5 for SSTA and between -0.3 and -0.5 for MHW frequency (Figures 10e and f). This correlation was strongest in the NRS and in the offshore areas of the central and southern RS. To our knowledge, no previous study has examined the relationship between the EATL/WRUS index and SSTA or MHW frequency in the RS. However, a recent study by Hamdeno and Alvera-Azcaráte (2023) reported a negative correlation between the EATL/WRUS index and SSTA in the eastern Mediterranean region, which is geographically close to the RS and is frequently affected by systems of Mediterranean origin (Langodan et al., 2017a, 2017b). The NAO index showed a negative correlation with the SSTA in the RS, with a correlation coefficient of less than -0.3 (Figure 10g). The influence of the NAO on the SSTA was stronger in the NRS than in the SRS. The correlation between the NAO index and MHW frequency was not significant, except in the SRS, where a positive correlation of 0.2 was observed, and in the Gulfs of Suez and Aqaba, where a negative correlation of -0.2 was observed (Figure 10h). The ONI showed a negative correlation with the SSTA in the RS, which was more pronounced in the coastal areas of the central and southern RS (Figure 10i). The correlation between the ONI and MHW frequency was not significant, except for a correlation of about -0.2 on the western coast of the SRS (Figure 10j). Overall, the AMO and IOD indices showed the strongest and most consistent correlations with SSTA and MHW frequency in the RS, while the EATL/WRUS index showed a strong negative correlation with both SSTA and MHW frequency, particularly in the NRS. The NAO and ONI indices showed weaker and less significant correlations with SSTA and MHW frequency in the RS.

The time series of various climate modes were compared with the annual frequency of MHWs in the RS as presented in Figure 11. The results revealed a positive correlation between the AMO and the annual frequency of MHWs in the RS. The years with the lowest MHW frequency, from 1982 to 1994, were found to align with the negative phase of the AMO. Conversely,

the years with the highest MHW frequency corresponded with the positive phase of the AMO, as shown in Figure 11a.

Furthermore, IOD was also found to have a significant relationship with the annual frequency of MHWs in the RS. From 1982 to 1993, the IOD had on average negative phase, and the frequency of MHWs was the lowest. In contrast, the last seven years of the study period, from 2015 to 2021, had high MHW frequencies, which corresponded with the positive phase of the IOD, as depicted in Figure 11b. The analysis also revealed a negative correlation between the EATL/WRUS index and the frequency of MHWs in the RS. The years with low MHW frequency were found to correspond with the positive phase of the

EATL/WRUS index, while the years with high MHW frequency corresponded with the negative phase of the index, as shown in Figure 11c. However, the comparison between the NAO and the ONI with the annual time series of MHW frequency did not suggest a clear relationship between their occurrence. The correlation maps in Figures 10e-h also support this finding, as the correlation was not significant in most parts of the RS. In conclusion, these results suggest that the AMO and IOD are the primary climate modes that align with the interannual variability of the MHWs frequency in the RS. The EATL/WRUS index

also has a significant relationship with the annual MHW frequency, while the NAO and ONI do not appear to have a substantial relationship.

In the last four decades, the year 2010 showed the highest MHW frequency particularly in the NRS, and in the same year had one of the strongest positive AMO phases (i.e. the AMO is positively correlated with the MHW frequency), while at the same time EATL/WRUS and NAO had their strongest negative phases during the entire study period.

**3.4 Case Study: 2010 MHWs in the Northern Red Sea**

The selection of 2010 as a case study for MHWs in the northern RS is based on several reasons. Firstly, 2010 was one of the warmest years on record, with a high frequency of MHWs in the region. Secondly, the spatial distribution of SSTAs and MHWDs in 2010 was found to be different from that of other warm years. Thirdly, although the SRS is known to be warmer than the NRS throughout the year (Fig. 2), in 2010 the SSTA of the NRS was higher by more than 1°C than the SRS

(Supplementary Figures S2. d). Therefore, this section aims to provide a detailed description of the spatial and vertical extent as well as the potential atmospheric drivers of the intense MHW event that occurred in the NRS in 2010.

During both winter and summer of 2010, the NRS experienced ten MHW events (Fig. 12). These included one severe event in February and March (Category III), one strong event between October and December (Category II), and several moderate events (Category I). In this section, we will provide a detailed analysis of the most intense and longest winter MHW event that

occurred in the NRS. This event occurred between February and March with an SSTA of about 4°C above the climatological average, peaking on March 12, 2010 (Figure 12). This event was categorized as severe (i.e. the temperature exceeded 3X the threshold). To better understand the dynamics of the event, the anomaly of water column temperature compared to the mixed layer depth (MLD) and time series of the MHW at different depth levels (surface, 25 m, 55 m, 110 m and 130 m) was calculated, as shown in Figure 13. The results showed that the regular diurnal cycle of water temperature, characterized by high

temperatures during the day and low temperatures during the night, gradually disappeared during the days of the MHW and was completely absent during the peak days of the event. This suggests that the ocean temperature reached a threshold where

the usual nighttime cooling was insufficient to lower the high SSTA and the heat was stored in the water column for the duration of the event. In addition, the results revealed a strong negative relationship between upper layer temperature and MLD, with a thin mixed layer aligned with the days of the higher water temperature. The time lag between the drop in MLD and the high water temperature was approximately four days, indicating that the sharp drop in MLD may have contributed to the rise of the water temperatures (Fig. 13a). The results also showed that the temperature anomaly extended vertically into the water column, reaching a depth of approximately 120 m during the MHW event (Fig. 13b-f). The duration of the MHW event varied from the surface to the subsurface, with the surface event (February 9 to March 18) being shorter than the events at 25 m and 55 m depth (February 9 to March 31), meaning that the heat from the MHW event was stored longer in the middle layer than at the surface. At 110 m depth, the duration of the MHW was shortest compared to the upper layers (February 26 to March 17) and occurred around the peak day of the surface event (March 12).

To better understand how atmospheric forcings may have contributed to the development of this MHW event, the spatial averages of atmospheric variables before (February 3 to 7), during and after (March 10 to 15) the event were calculated and presented in Figure 14. Additionally, the time series of atmospheric variables averaged over the NRS (24° - 28° N and 34° - 39° E) during the event are presented in Supplementary Figure S8. Prior to the MHW event, the average SSTA in the NRS was about 1°C above average, while it was negative in the SRS and in the Strait of Bab El-Mandab. During the MHW event, the SSTA increased in the NRS and reached a local maximum of 4°C above the climatological average (Figure 14a-c). The spatial distribution of the average Tair showed higher values in the west (over Egypt, Eritrea and Ethiopia) than in the east (over Saudi Arabia) (Figure 14d-f). Over the NRS, the Tair increased by approximately 8°C compared to before the event. After the MHW, the Tair decreased but did not return to pre-MHW values (Figure 14d-f and Figure S8b). The MSLP maps showed an opposite distribution to Tair, with areas of high Tair having low MSLP and vice versa (Figure 14j-l). In addition, the average MSLP over the NRS decreased during the MHW event compared to before the event (Figure S8c). Before the MHW event, the winds blew from the eastern region and mainly flowed towards the SRS. During this event, the winds blew from the south and shifted to the west before reaching the NRS region, which experienced very low winds (Figures 14m-o and Figure S8d). Furthermore, the relative humidity rose by 10% over the NRS during the MHW period (Figure S8e).

In the RS, the latent heat flux (LHF) shares a similar spatial and temporal distribution with the Qt (Nagy et al., 2021). The majority of the net surface heat exchange variability in the NRS is known to depend on the turbulent components of the surface flux, primarily the LHF (Papadopoulos et al., 2013). In our case study, before the MHW event, the LHF ranged from -140 to -60 W/m², and the Qt ranged from -150 to -20 W/m², indicating that the ocean was losing heat to the atmosphere (Figure 14g-i and Figure S8f).

During the MHW, the combined effect of increased Tair, humidity and reduced winds led to a strong decrease in the ocean latent heat loss, signifying reduced heat loss to the atmosphere. Particularly during the days of the MHW onset and peak, the LHF fluctuated between -20 and -10 W/m². This decrease coincided with a slight increase in net solar radiation from 180 W/m² before the MHW to more than 200 W/m² during the MHW (Figure S8f). Accordingly, the heat exchange between the air and

440 ocean reversed, causing a prolonged ocean heat gain, with Qt reaching up to 100 W/m², ultimately driving the MHW (Figure 14g-i).

In summary, our findings indicate that the late winter MHW event in the NRS was primarily driven by atmospheric forcing, specifically an increase in Tair and humidity, possibly linked to reduced winds. These atmospheric conditions collectively resulted in reduced LHF and a strong ocean heat gain, creating favourable conditions for MHW occurrence.

**4. Conclusions**

This study aimed to analyse the characteristics and trends of SST and MHWs in the RS and investigate their relationship with climate modes. Over the past four decades, the RS has experienced a significant increase in SST and MHWs, with a notable acceleration in the past decade. The spatial distribution of MHWs showed high variability, with the highest frequency in the coastal areas of the southern region and the Strait of Bab El-Mandab. The mean duration of MHWs was longer in the northern
region and the Gulfs of Suez and Aqaba, while the most intense MHWs were observed in the northern region and the western part of the southern region. The study revealed a warming trend in the RS since the mid-1990s, with a notable increase after 2016. A total of 78 MHW events and 1016 MHWDs were identified in the RS over the past four decades, with 46% of these events and 58% of the days occurring in the last decade. Considering the results of this study and the observed trends of MHWs in the region, it is recommended that future work considers an analysis of MHW trends based on different baselines. This
comparison is particularly important when it comes to projecting future MHWs under different global warming scenarios, as the selection of an appropriate baseline is of utmost importance for the detection of future MHWs and the calculation of their trends.

The analysis of SSTA trends revealed a decadal variability with high trends alternating between the NRS and SRS. From 1982 to 1991, the highest trends were observed in the NRS, with an average trend of 0.56°C/Decade. However, from 1992 to 2001,
the spatial pattern of the SSTA trend was altered, with the highest trends observed in the SRS, with an average trend of 0.57°C/Decade, and lower trends in the NRS, with an average trend of 0.30°C/Decade. From 2002 to 2011, the highest trends were again observed in the NRS, with an average trend of 0.45°C/Decade, while the SRS experienced no trend in the SSTA during this period. Finally, over the last decade of the study period (2012-2021), the SRS had higher trends in the SSTA than the NRS, with an average trend of 1.35°C/Decade for the SRS and 0.89°C/Decade for the NRS. The spatial distribution of
average SSTA and MHWDs was analysed in both warm and cold years. In cold years, the NRS and the Strait of Bab El-Mandab had the highest SSTA and MHWDs. However, in warm years, the SRS had the highest SSTA values, and the SRS and the northern regions of the Gulfs of Suez and Aqaba had the highest number of MHWDs the year of 2010 an exception among the warm years, with the northern region having the highest SSTA and MHWDs.

Investigating the relationship between climate modes and SSTA and MHW frequency, it was found that the AMO and the
470 IOD indices had high positive correlations with SST and MHW frequency in the RS. Meanwhile, the EATL/WRUS index showed a negative correlation with both SST and MHW frequency, particularly in the northern region. The NAO and the ONI

indices showed weaker and less significant correlations with SST and MHW frequency in the RS. The study further examined the intense MHW event that occurred in the northern region between February and March of 2010. This MHW has extended to 120 m depth and was associated with a combination of atmospheric conditions, specifically an increase in Tair and humidity which were possibly linked to the reduced winds and resulted in reduced LHF and a strong ocean heat gain, creating favourable conditions for MHW occurrence. Interestingly, the AMO and the IOD were in a robust positive phase in 2010, while the EATL/WRUS and the NAO were in their most pronounced negative phase. These climate indices have been shown to be correlated with SSTs and MHWs in the RS, and their combination in 2010 may have contributed to the increased occurrence of MHWs in that year.

In conclusion, this study has provided valuable insights into the characteristics and trends of SST and MHWs in the RS and their relationship with climate modes. The findings of this study can be useful for the management and conservation of marine ecosystems in the RS, as well as for the prediction and mitigation of the impacts of MHWs on these ecosystems. This research on MHWs in the RS region will also enable the generation of new scientific knowledge and help to fill gaps in the existing literature and advance marine science. For future work, the compound between MHWs and other extreme events will be investigated, and their impact on the RS ecosystem will also be studied.

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

**Funding:**

This work benefits financial support of the Aspirant F.R.S.-FNRS (Fonds de la Recherche Scientifique de Belgique, Communauté Française de Belgique) through funding the position of AB and funding a Aspirant - ASP grant.

**Conflict of interest:**

The authors declare that the research was conducted in the absence of any commercial or financial relationships that could be construed as a potential conflict of interest. Some authors are members of the editorial board of journal Ocean Science.

**Figures:**

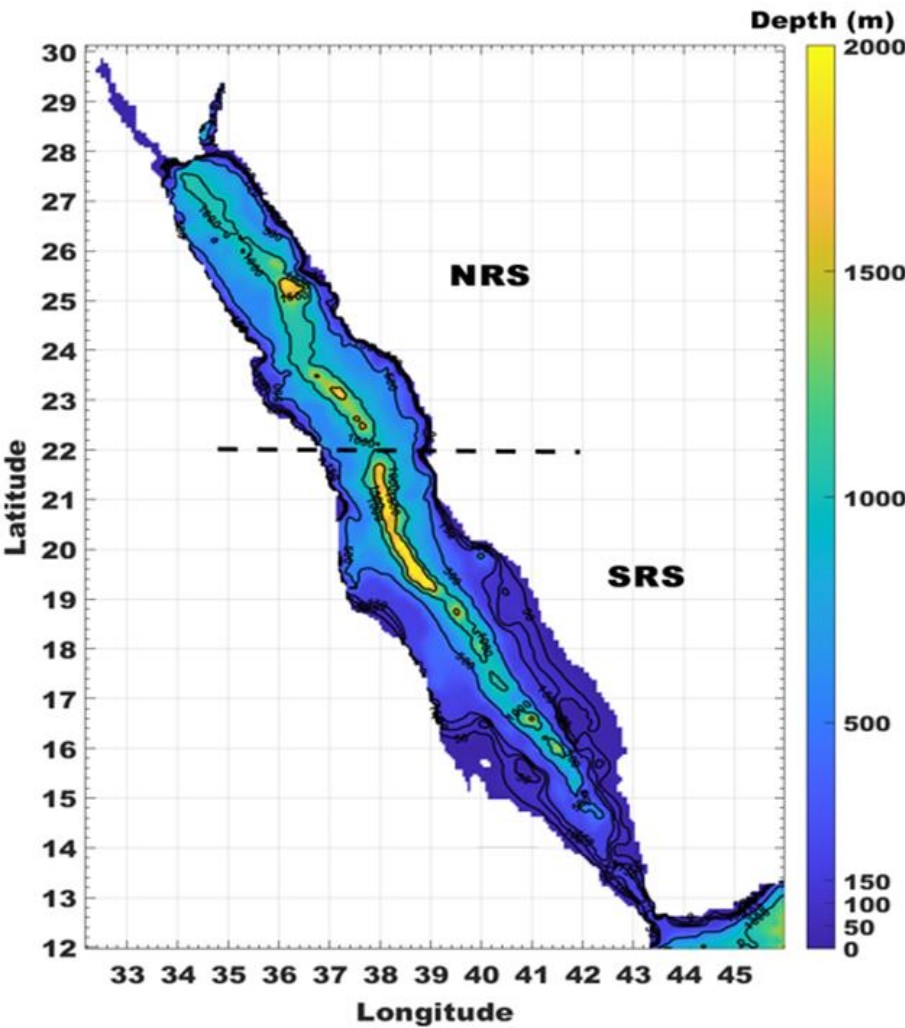

Figure 1. Bathymetry map of the Red Sea. Bathymetry corresponds to the GEBCO bathymetry dataset ( www.gebco.net).



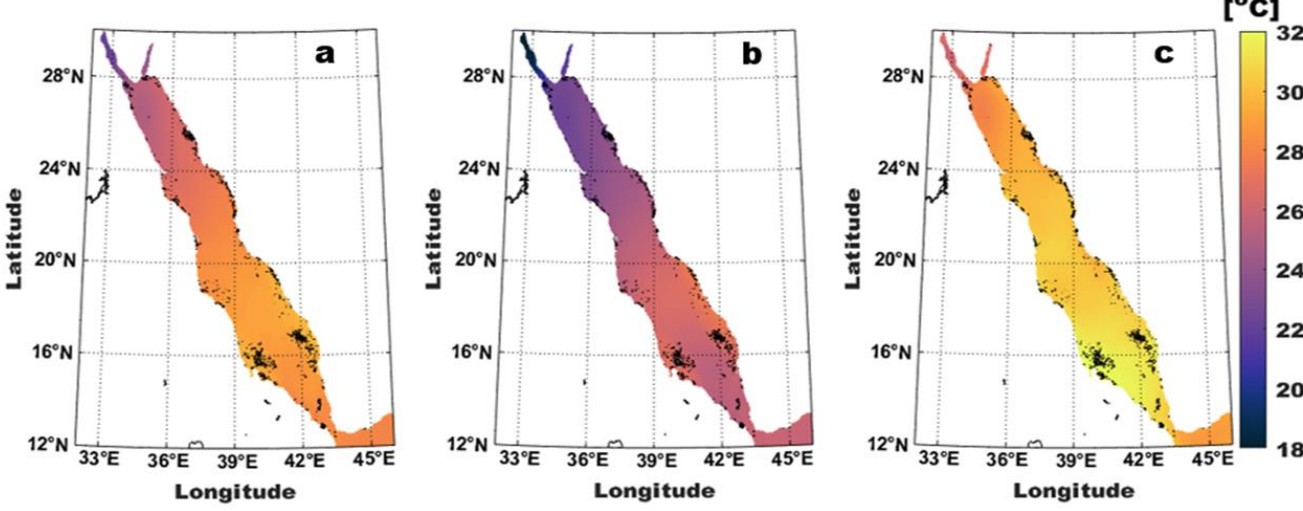


Figure 2. The spatial distribution of average Red Sea SST (in ºC) from 1982 to 2021. (a) over the entire study period, (b) during winter (January, February, and March), and (c) during summer (July, August, and September).




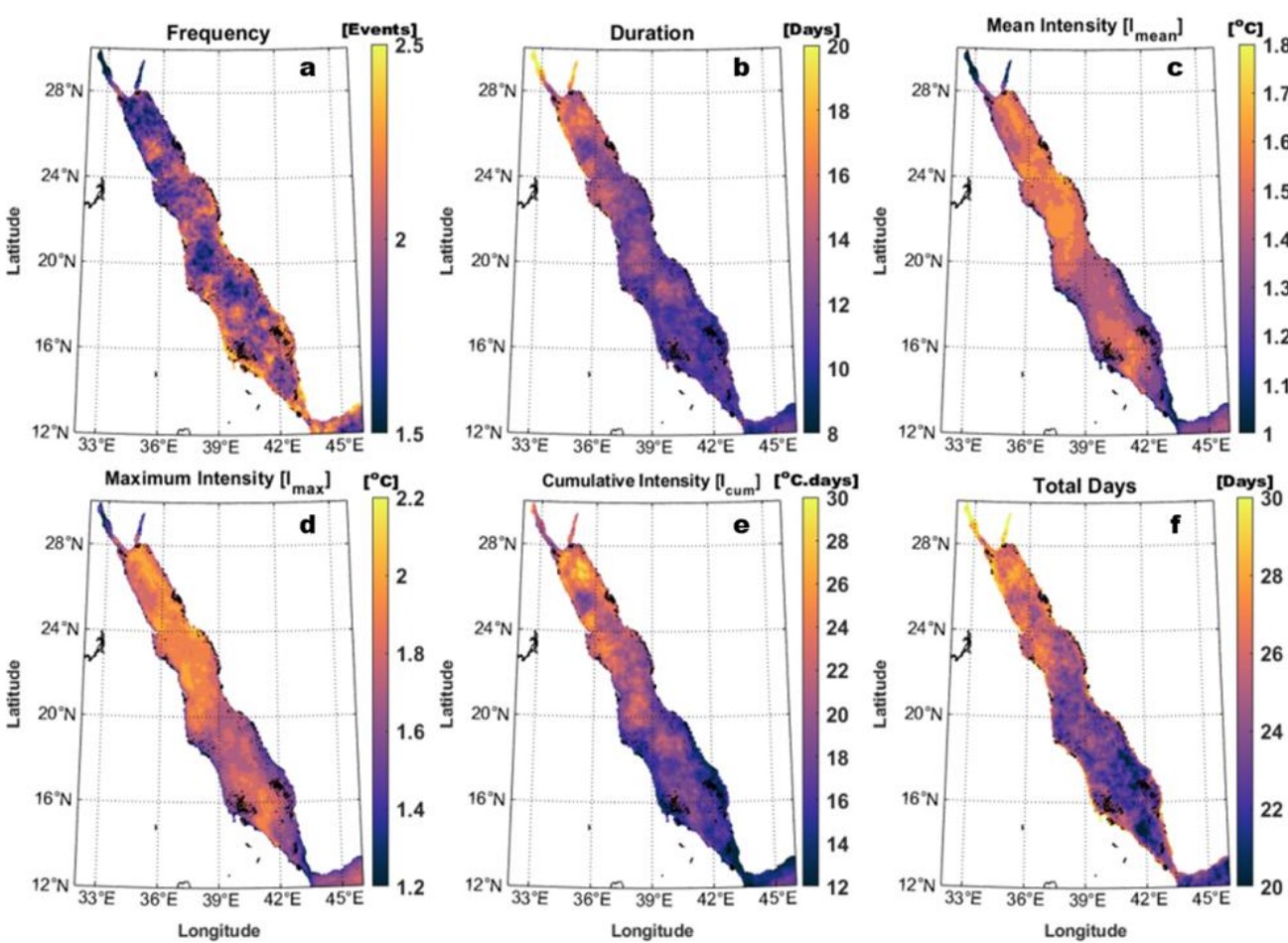


Figure 3. The spatial distribution of average MHWs characteristics at RS between 1982 and 2021. The average MHWs frequency (a), duration (b), mean intensity (c), maximum intensity (d), cumulative intensity (e), and total days (f).



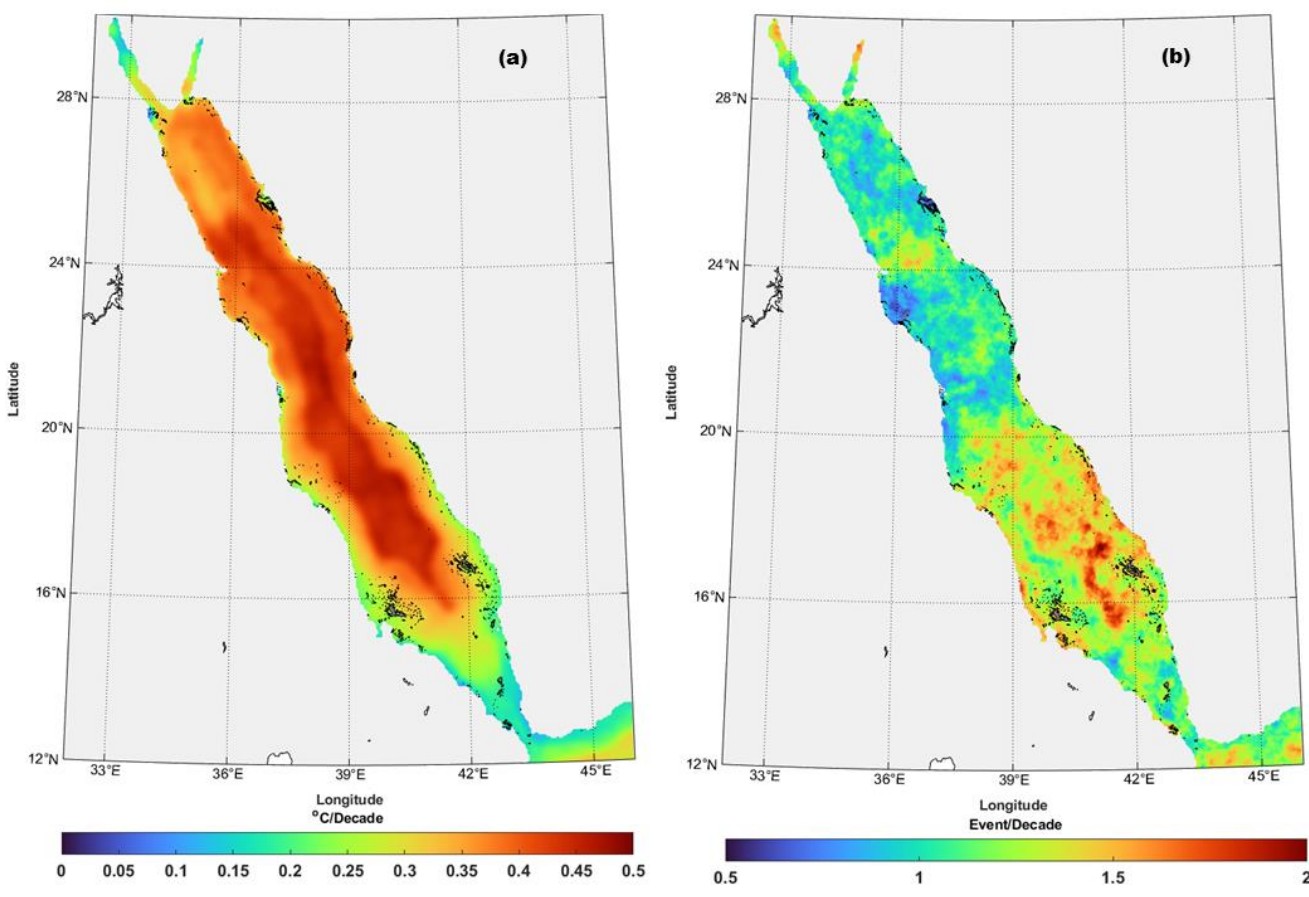


Figure 4. Trends in (a) sea surface temperature (°C/decade) and (b) marine heatwave frequency (events/decade) in the Red Sea from 1982 to 2021.



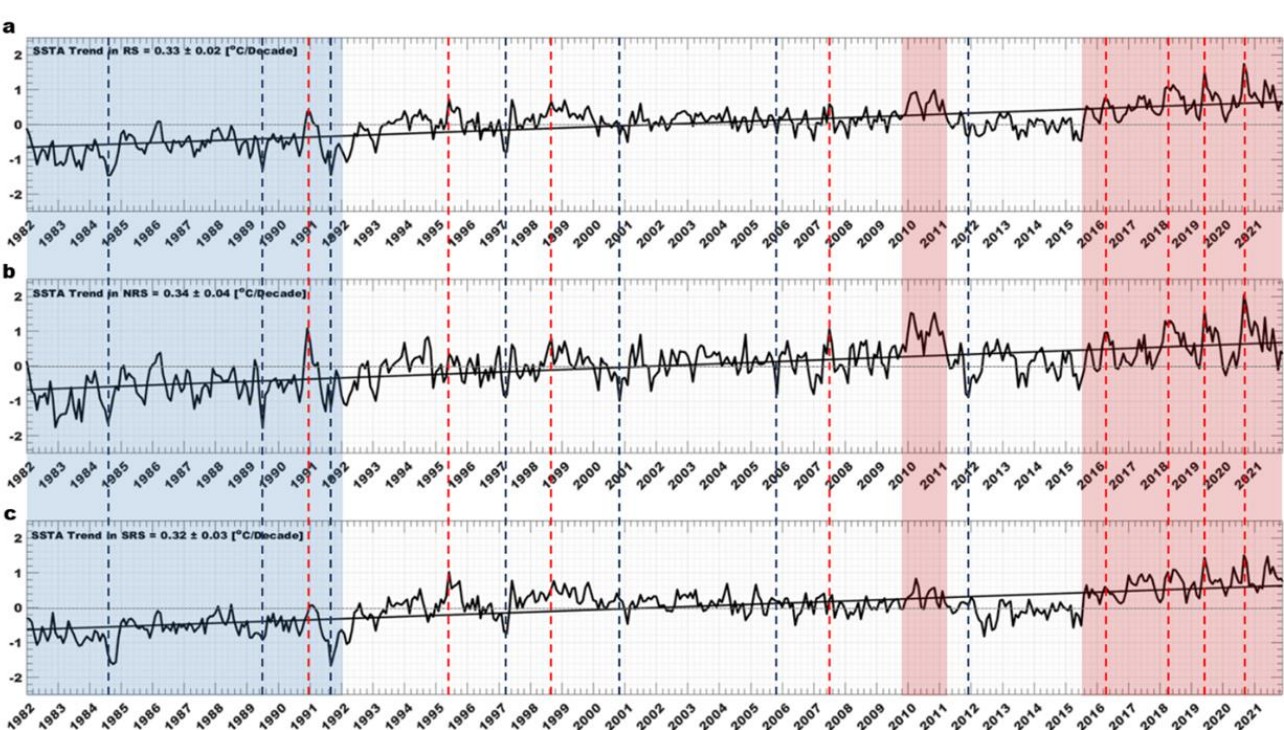

Figure 5. Temporal evolution and trend of sea surface temperature anomalies (°C) in the entire Red Sea (a), northern Red Sea (b), and southern Red Sea (c) from 1982 to 2021. The blue and red shaded areas represent the cold and warm periods, respectively. The blue and red dotted lines represent the coldest and warmest years, respectively.



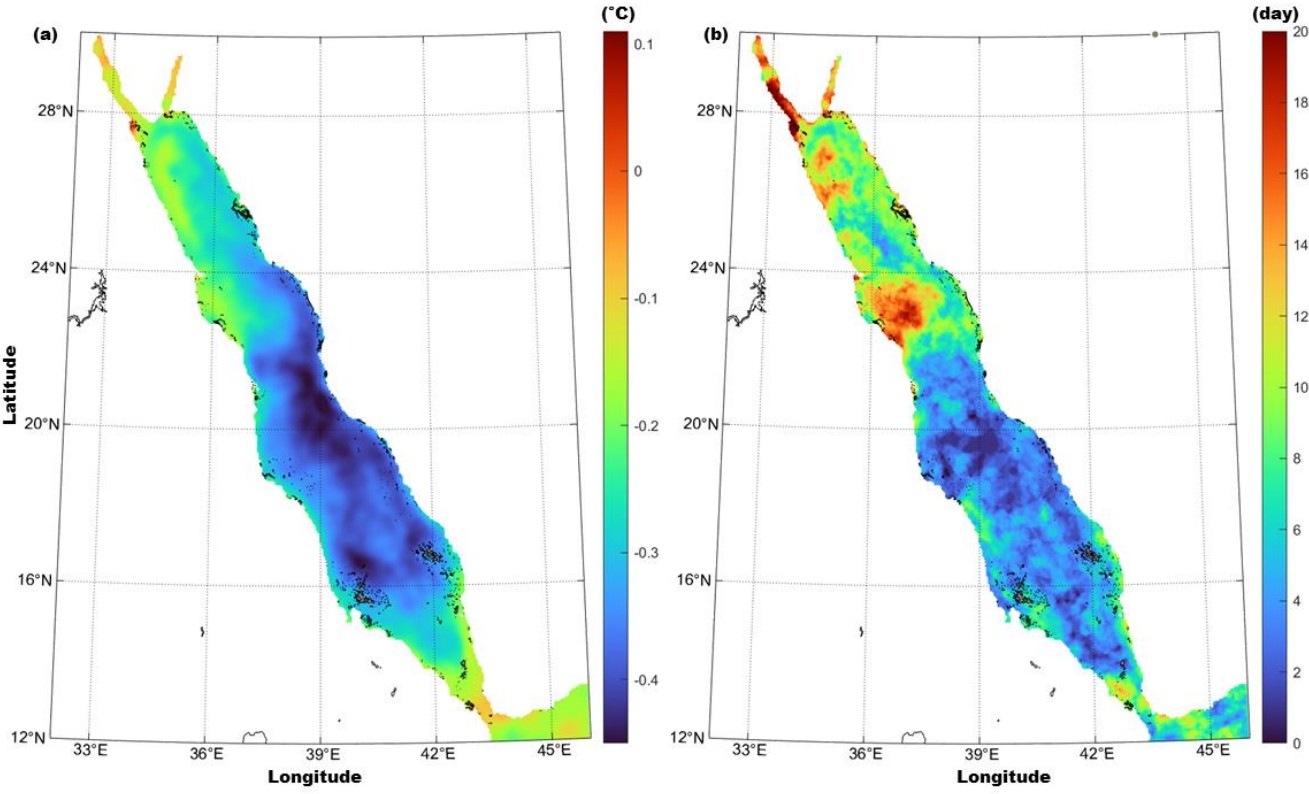

Figure 6. Spatial distribution of the average (a) sea surface temperature anomaly (°C) and (b) marine heatwave days (days) in the Red Sea during the coldest years of the study period (1982-2021).




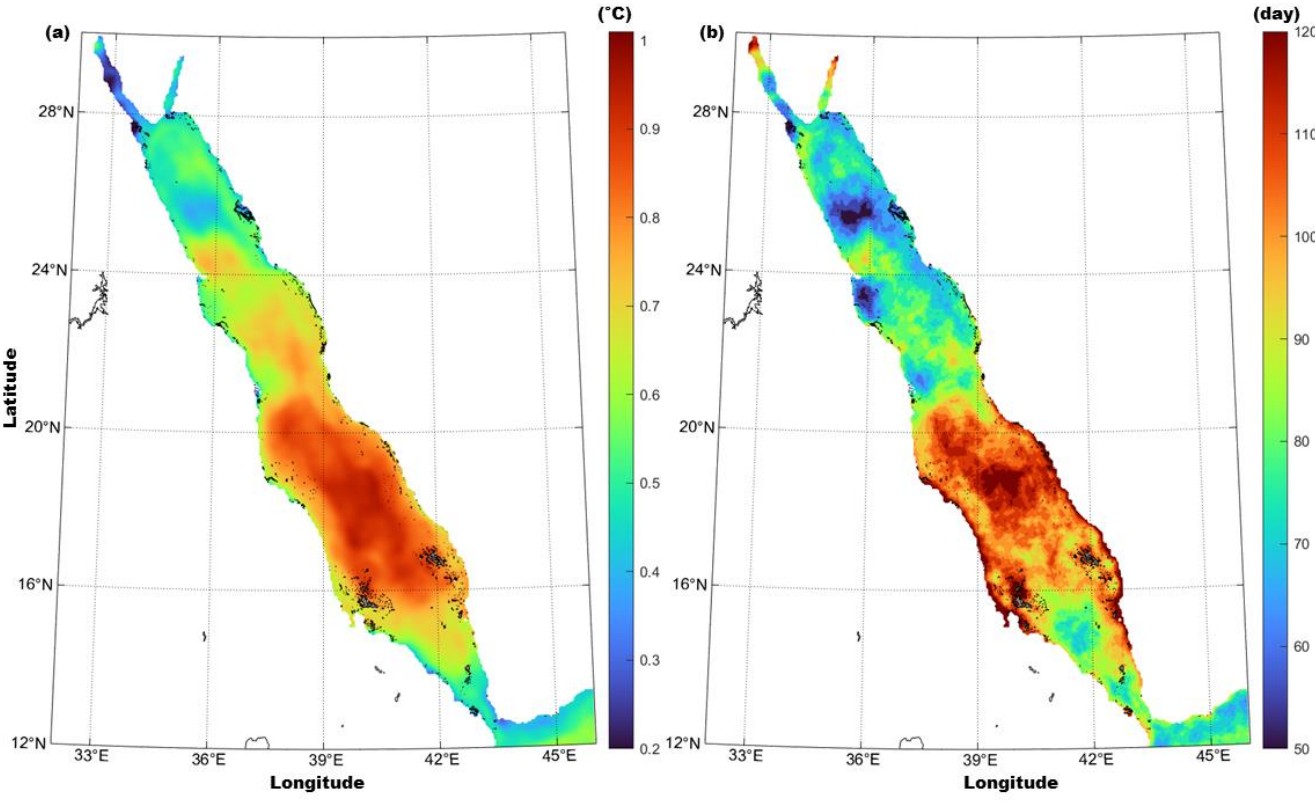


Figure 7. Spatial distribution of the average (a) sea surface temperature anomaly (°C) and (b) marine heatwave days (days) in the Red Sea during the warmest years of the study period (1982-2021).



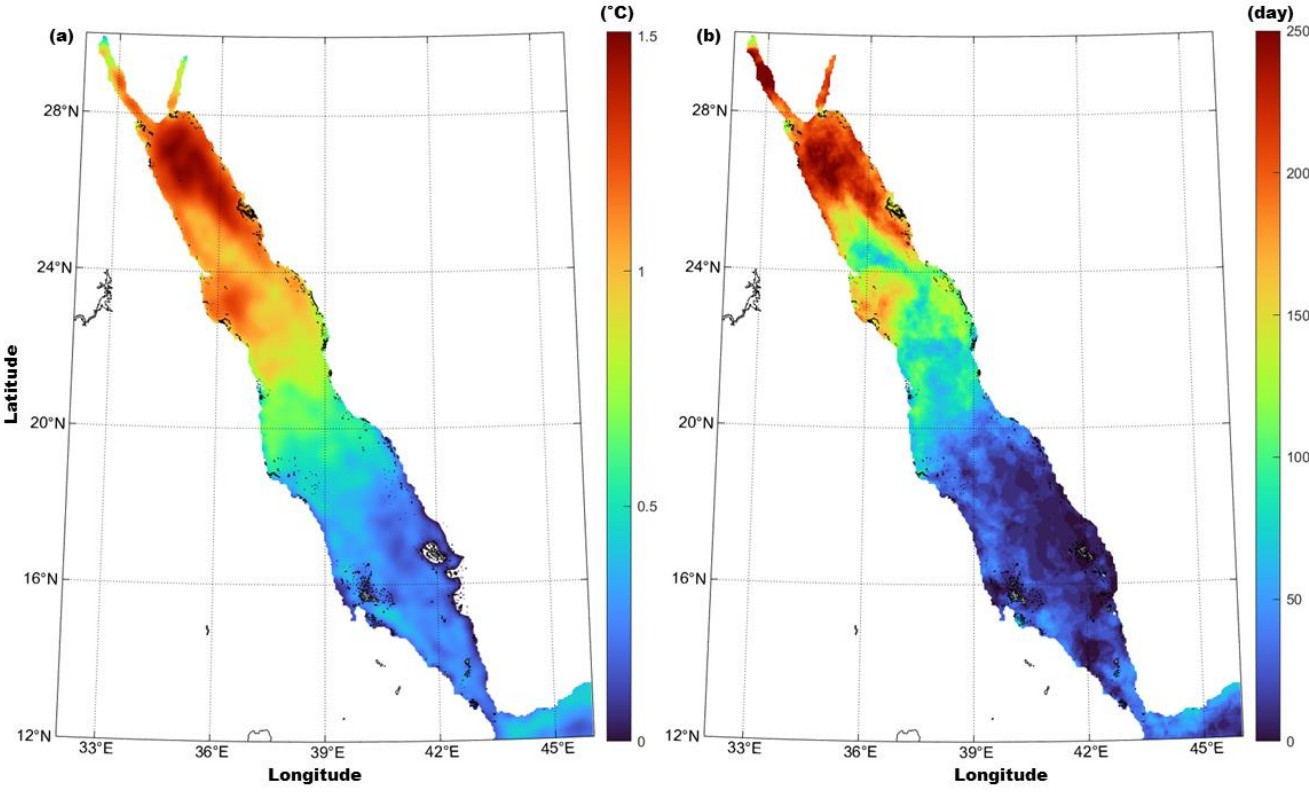


Figure 8. Spatial distribution of the average (a) sea surface temperature anomaly (°C) and (b) marine heatwave days (days) in the Red Sea during the year 2010.




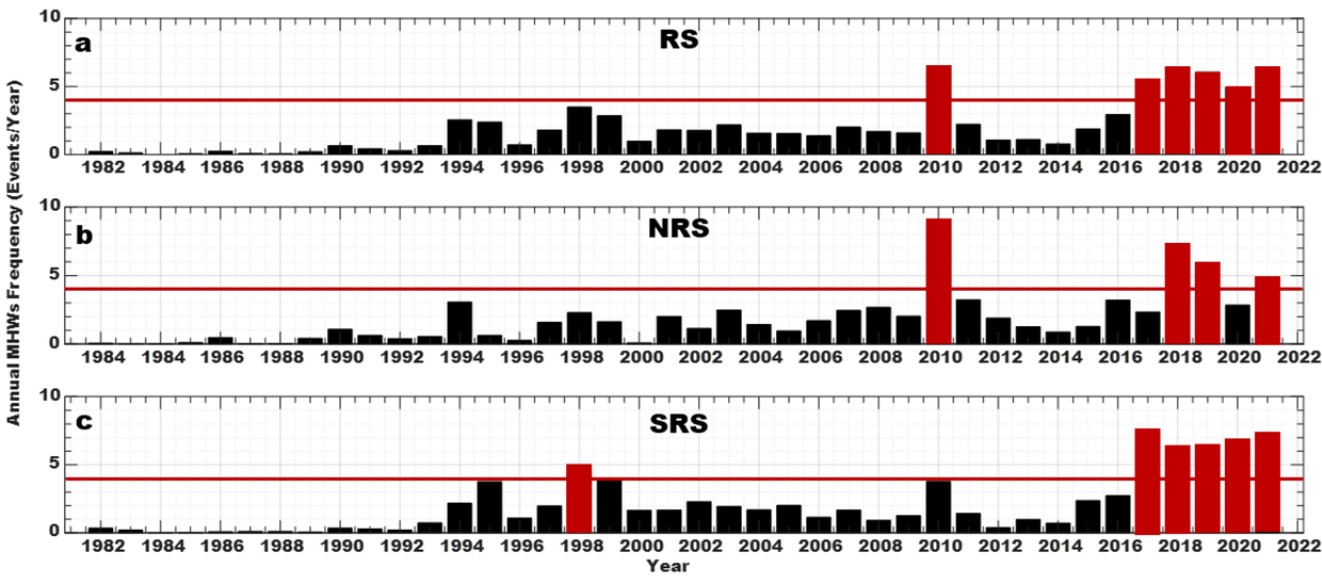

Figure 9. Interannual variability of marine heatwave frequency in the entire Red Sea (a), northern Red Sea (b), and southern Red Sea (c) from 1982 to 2021. The red bars represent the years with the highest frequency in each basin.




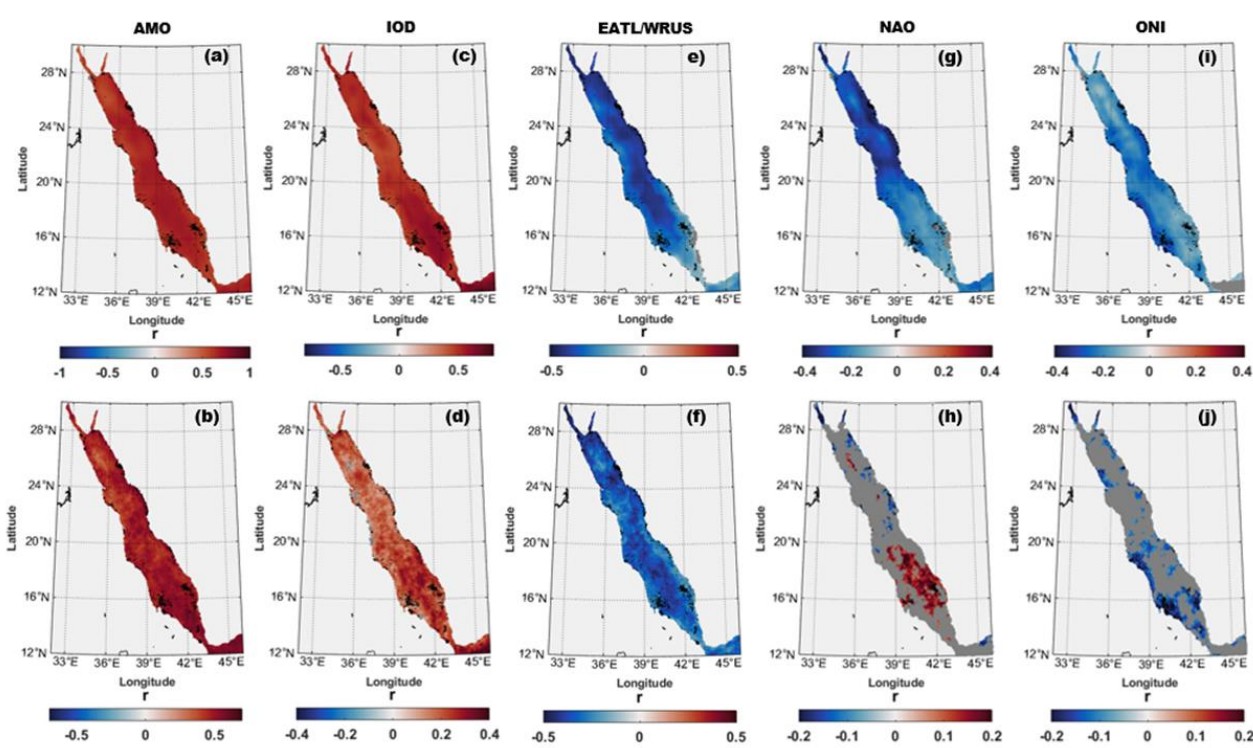

Figure 10. Correlation maps of the SST anomaly (upper panels) and marine heatwave frequency (lower panels) in the Red Sea with different climate modes from 1982 to 2020. Correlations are shown with the AMO index (a, b), IOD pattern (c, d), EATL/WRUS index (e, f), NAO index (g, h), and ONI index (i, j). Gray shading indicates areas where the correlation is not significant (p-value > 0.05).



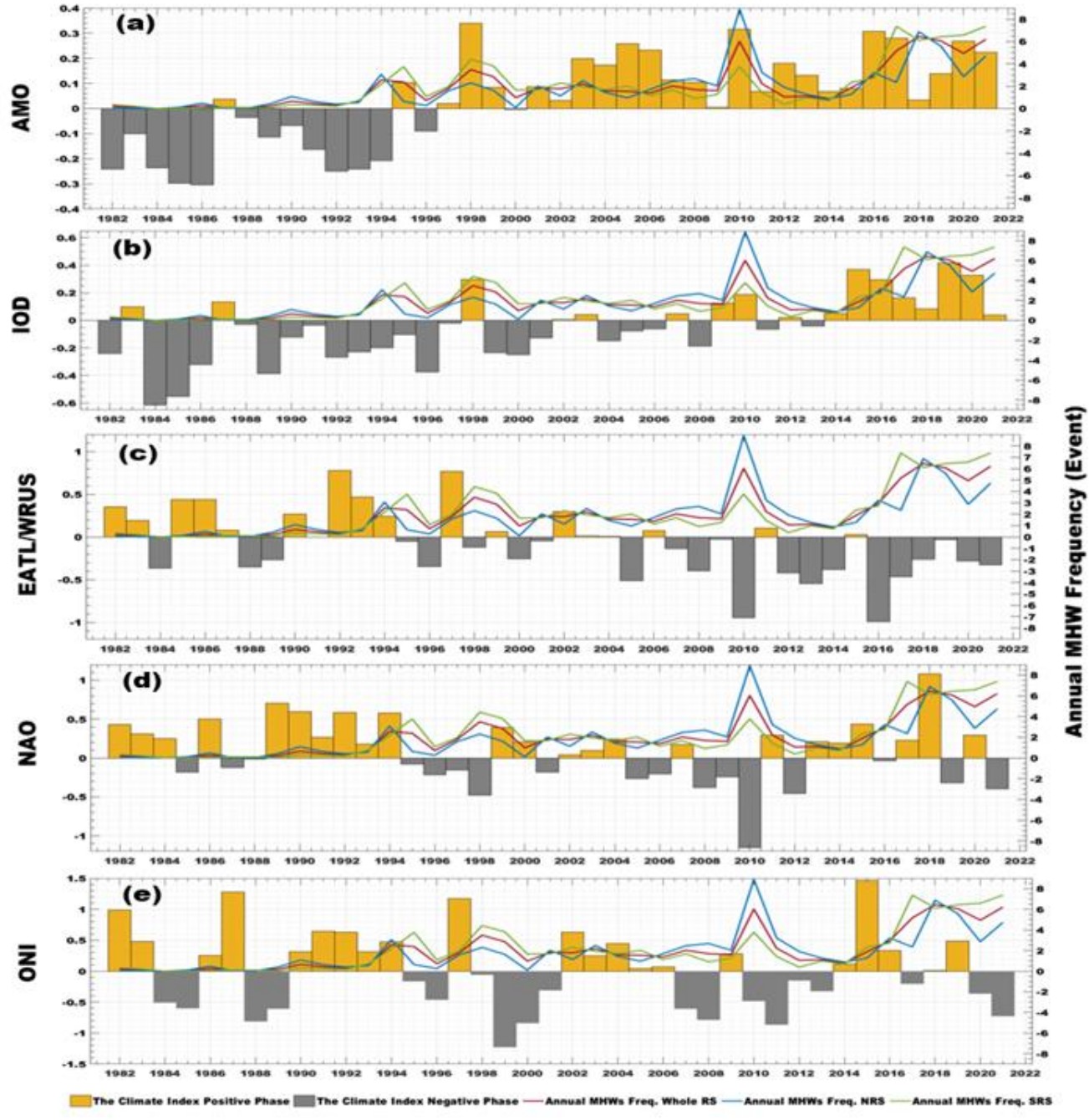


Figure 11. Annual time series of normalized climate indices and marine heatwave frequency in the Red Sea. (a) AMO index, (b) IOD index, (c) EATL/WRUS pattern, (d) NAO index, and (e) ONI index are shown with annual MHW frequency in the entire Red Sea (red line), northern Red Sea (blue line), and southern Red Sea (green line). Yellow and gray bars indicate the positive and negative phases of the climate indices, respectively.

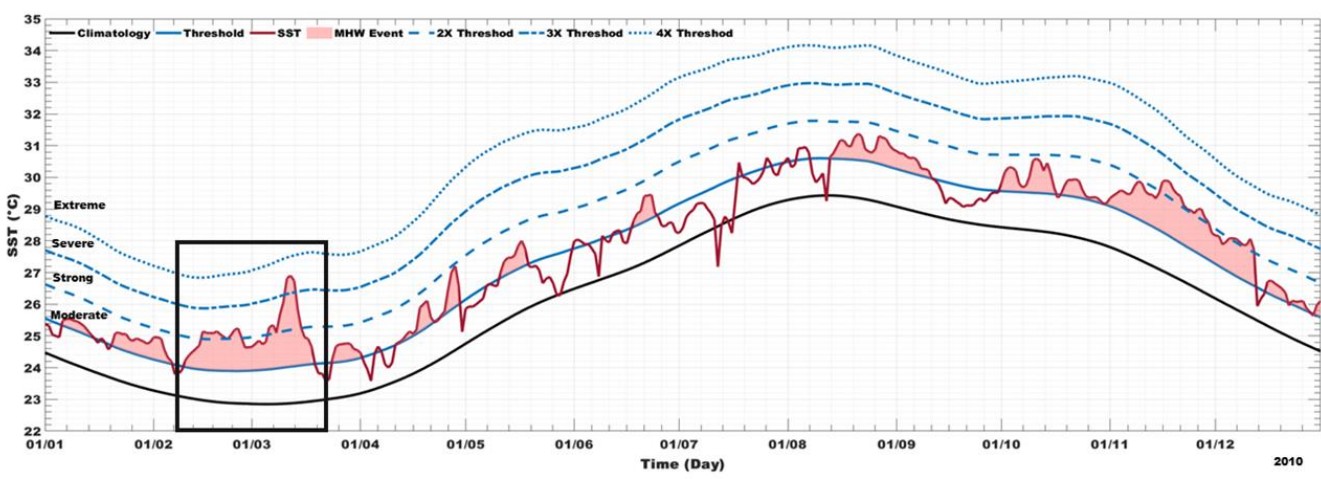


Figure 12. Marine heatwave events in the northern Red Sea in 2010. The red shaded area indicates the duration of the event. The solid red line represents the sea surface temperature, the solid black line represents the climatology, the solid blue line represents the threshold, and the dotted blue lines represent the multiples of the threshold (defining the MHW categories).


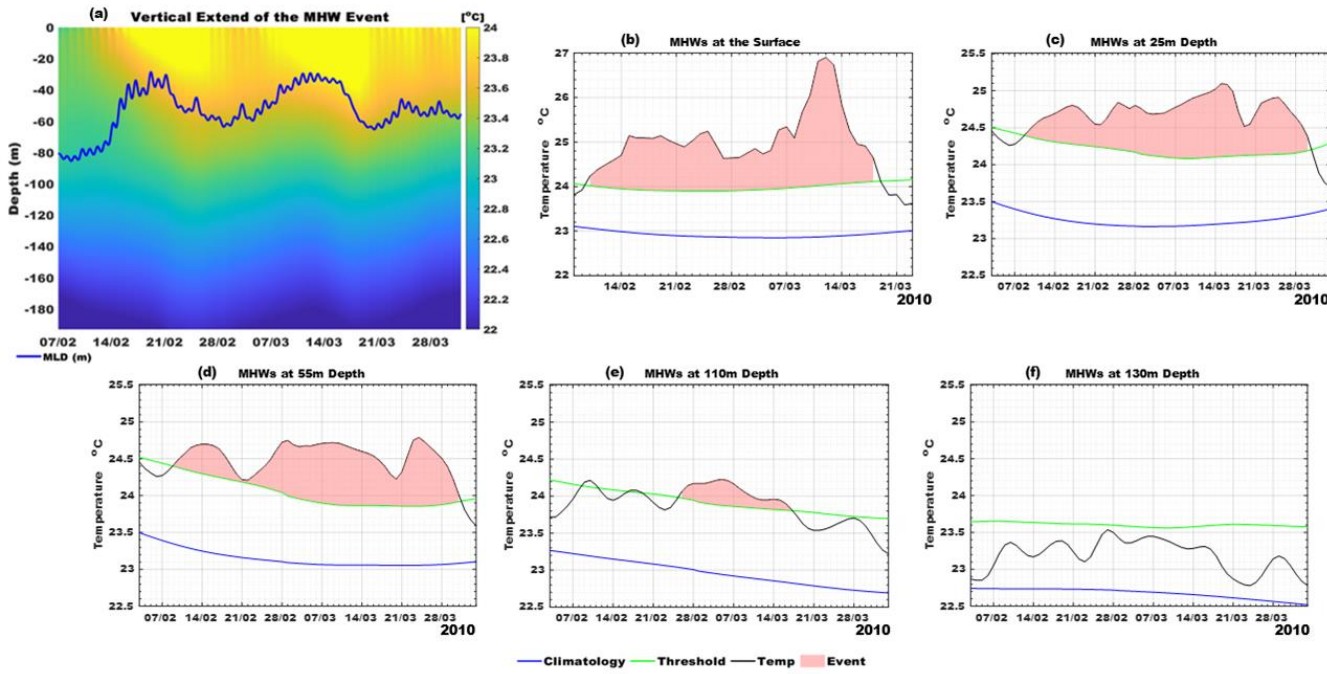


Figure 13. The marine heatwave (MHW) event in the NRS between February and March 2010. (a) The vertical extent of the MHW, with the blue line representing the mixed layer depth (MLD). (b-f) The MHW at different water column depths (surface, 25m, 55m, 110m, and 130m), with the red shaded area indicating the MHW event, the black solid line representing the sea

surface temperature (SST), the blue solid line representing the climatology mean, and the green solid line representing the 90th percentile threshold.



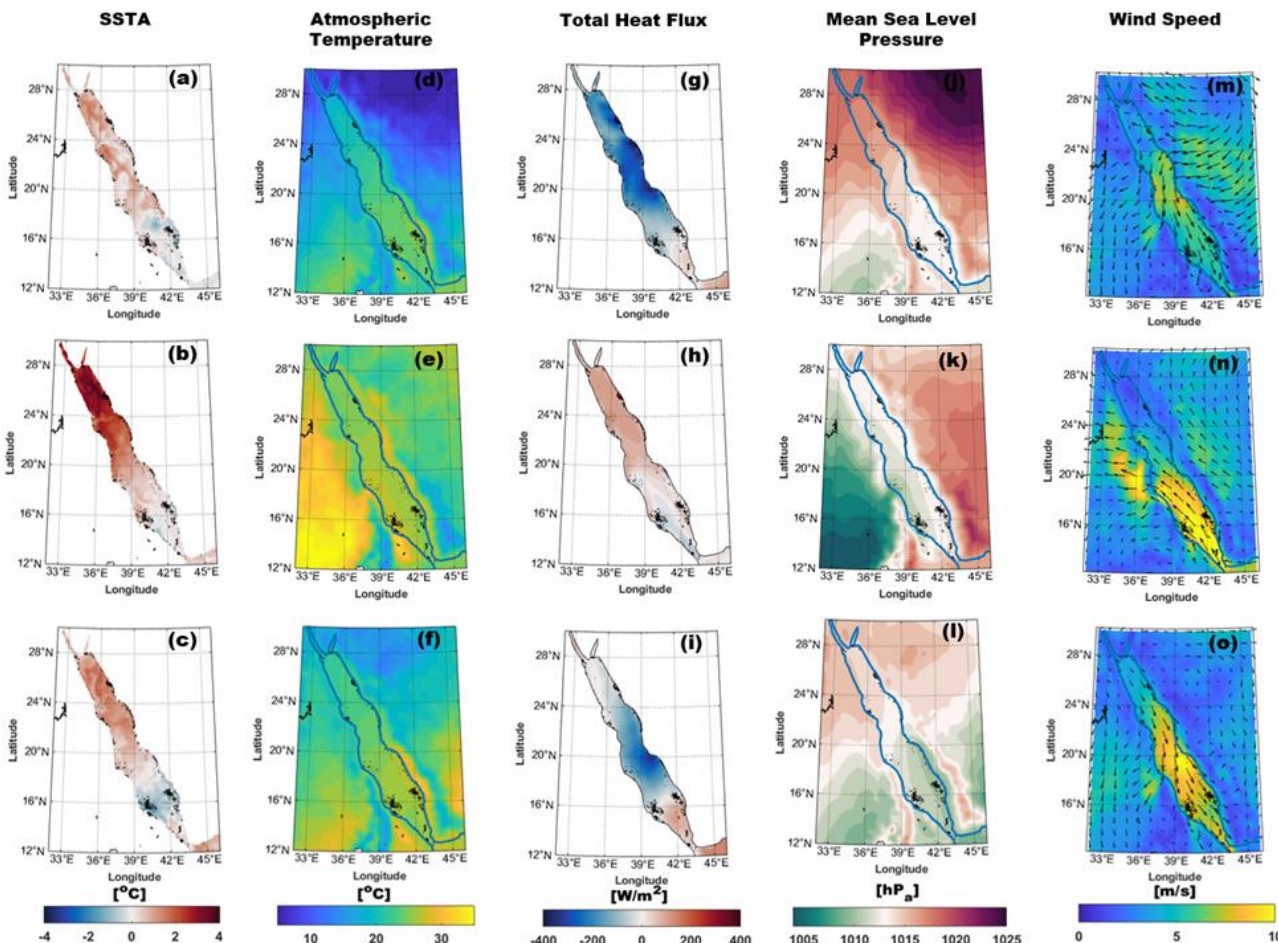

Figure 14. The average spatial distribution of atmospheric variables before, during, and after the marine heatwave event in the NRS. The upper panels show the period before the MHW event (3rd to 7th of February), the middle panels show the period during the MHW event (10th to 15th of March), and the lower panels show the period after the MHW event (20th to 25th of March). Panels (a-c) represent sea surface temperature anomaly (SSTA, in °C), panels (d-f) represent atmospheric temperature (in °C), panels (g-i) represent total heat flux (in W/m2), panels (j-l) represent mean sea level pressure (in hPa), and panels (m-o) represent wind speed (in m/s) and wind direction.