# Peer review of "Investigating the Long-term Variability of the Red Sea Marine Heatwaves and their Relationship to Different Climate Modes: Focus on 2010 Events in the Northern Basin"

_EGUsphere, 2024_

## Author Comment (AC1)

Review of Marine Heatwaves in the Red Sea and their Relationship to Different Climate Modes: A Case Study of the 2010 Events in the Northern Red Sea

Main comments

In this paper, the authors have run an extensive analysis of SST and MHW events in the Red Sea and also at a regional level (North and South RS). They have characterized SST anomaly and MHWs and also assessed long-term trends. Additionally, an attempt to relate atmospheric variables in the region with MHWs was run and a winter event on 2010 was analysed with higher detail.

The work presented in the manuscript is certainly of interest, especially in an area as unique as the Red Sea but some concerns arise from the text. Regarding the title of the manuscript, the case study of 2010 seems to be a main aim but the discussion in the corresponding section is not as extensive as the reader could expect. I suggest changing the title or deepening the case study analysis. Besides, I could not find a proper justification of the event studied. Was it a record event? The most intense? Which is the interest of studying this event?

**We would like to express our gratitude to the Reviewer for his/her insightful comment and valuable feedback on our paper. We have carefully considered all the Reviewer suggestions in the revised Manuscript. Detailed point-by-point reply to all their questions are given below.**

**In response to the suggestion to change the title, we propose "Investigating the Long-term Variability of the Red Sea Marine Heatwaves and their Relationship to Different Climate Modes: Focus on 2010 Events in the Northern Basin" to emphasize the focus on the 2010 events.**

**Regarding the justification for studying this particular event, it was the most intense event of that year, longest winter event and occurred during the winter season, making it an interesting case to investigate its potential drivers. We have added new texts in the Introduction and Discussion Sections to provide further explanation for the selection of 2010 events as a case study.**

**Lines (119-121): "2010 was selected as a case study as it was one of the warmest years with highly frequent MHWs and had a different spatial distribution of SSTA and marine heatwave days (MHWDs) than the other warm years."**

**Lines (389-398): "The selection of 2010 as a case study for MHWs in the northern Red Sea is based on several reasons. Firstly, 2010 was one of the warmest years on record, with a high frequency of MHWs in the region. Secondly, the spatial distribution of SSTAs and MHWDs in 2010 was found to be different from that of other warm years. Thirdly, although the SRS is known to be warmer than the NRS through out the year (Fig. 2), in 2010 the SSTA of the NRS was higher by more than 1°C than the SRS (Supplementary Figures S2. d). Therefore, this section aims to provide a detailed description of the spatial and vertical extent as well as the potential atmospheric drivers of the intense MHW event that occurred in the NRS in 2010.**

**During both winter and summer of 2010, the NRS experienced ten MHW events (Fig. 12). These included one severe event in February and March (Category III), one strong event between October and December (Category II), and several moderate events (Category I). In this section, we will provide a detailed analysis of the most intense and longest winter MHW event that occurred in the NRS."**

Another main concern is that the analysis of the relation between atmospheric variables and MHWs, or SSTA, seems too cursory despite the undoubted interest it may have. I recommend that the authors describe this analysis in more detail, as I assume that this is work that has already been done but it is not sufficiently highlighted in the text.

**We thank the Reviewer for pointing this out, this helped us to improve the Methodology Section. A paragraph was added in the methodology to describe in detail the analysis of the relation between the climate modes and MHWs/SSTA, as follows:**

**"To gain a deeper understanding of the relationship between the different climate modes and the occurrence of MHWs over the last four decades in the RS, spatial correlations were examined. The climate modes considered in this study are the Oceanic Niño Index (ONI), the East Atlantic/West Russia Pattern (EATL/WRUS), the Atlantic Multidecadal Oscillation (AMO), the North Atlantic Oscillation (NAO) and the Indian Ocean Dipole (IOD). The correlation maps were calculated using the Pearson correlation coefficient (r), a widely used method for measuring linear correlations between two variables (Kirch, 2008; Patten and Newhart, 2017). The Pearson correlation coefficient ranges from -1 to 1, where -1 stands for a perfect negative correlation, 1 indicates a perfect positive correlation and 0 for no correlation. In this study, we calculated the correlation maps between the annual time series of each climate mode and the annual MHWs/SSTA in the RS. The MHWs were identified and characterized using a set of metrics, such as their duration, frequency, mean intensity, maximum intensity, cumulative intensity and total days (as described in the previous Section). We calculated the correlation between the different annual climate modes and annual MHW metrics, in particular frequency, duration and total days. The results showed a consistent spatial correlation pattern with MHW different metrics, while the correlation coefficients varied only slightly. For the sake of brevity, we present only the correlation results with MHW frequency and SSTA in our results. The MHW frequency was chosen for presentation due to its slightly higher correlation compared to MHW duration and total days. To test the significance of the correlations, a two-tailed t-test was used (Patten and Newhart, 2017). The t-test is a statistical hypothesis test that compares the means of two samples and determines whether they differ significantly from each other. Finally, we also compared the time series between different climate modes and the frequency of MHWs in the RS and its sub-basins. By analyzing the correlation maps and the significance of the correlations, we can gain insights into the potential co-variability between MHWs in the RS and larger-scale climate variability."**

With this more detailed analysis and the other Reviewers' suggestions I would like to see a newly revised conclusions section to give some potentially interesting findings in the field of MHWs.

I would like to congratulate the authors on their work and encourage them to further develop it according to the indications received in this review process, prior to publication. My final decision is to review again after major revision but not because of problems in the methodology or conclusions but to clarify and deepen some of the analysis to improve the final result.

**The Authors once again would like to thank the Reviewers for their kind comments on this paper and their very valuable evaluation. We have taken all suggested comments into account and answered all question**

Minor comments

About Red Sea subregions, I could not find a justification for the spatial division between NRS and SRS. Please, indicate the spatial division criteria in the text.

**We agree with the Reviewer and a paragraph was added in the Methodology Section to explain the choices of the subregions, as follow:**

**"To provide a more comprehensive and detailed description of MHWs in the RS, we have divided the RS into two regions: the Northern Red Sea (NRS) and the Southern Red Sea (SRS). The NRS extends from 22°N to 30°N, while the SRS extends from 22°N to 12.5°N. This division was based on the north-south spatial thermal gradient in the RS, which shows different characteristics of SST and MHWs between the regions."**

Why do you consider winter months Jan-Feb-Mar? And summer months? It seems a little bit artificial division. Please better justify the Reviewer's selection or, better, consider different periods for winter and summer months.

**We appreciate the Reviewer's feedback and the opportunity to clarify our methodology. The selection of winter and summer months was based on the seasonal cycle of SST, with the three months of the lowest SSTs representing the winter season, and the three months of the highest SSTs representing the summer season. Our focus on these two seasons was intentional, as we observed that the most intense Red Sea MHWs occurred predominantly during winters and summers.**

**For more clarity, we have added a paragraph in the Methodology Section explaining the method of selecting the winter and summer months as follows:**

**"The winter and summer SST in the RS was calculated and averaged over the study period (1982-2021) at each grid point. The winter season was represented by the months of January, February, and March, while the summer season was represented by the months of July, August, and September. The selection of winter and summer months was based on the seasonal cycle of SST, with the three months of the lowest SSTs representing the winter season and the three months of the highest SSTs representing the summer season. We focused on these two seasons as it was observed that the most intense RS MHWs occurred predominantly during winters and summers."**

In your analysis, you mostly describe winter and summer months. What happens to SST and MHWs in spring and summer months? Sometimes you refer to annual frequencies, MHWDs… Please, be consistent with the periods selection and analysis or better indicate why and when such periods are being analysed.

**As mentioned in our response to the Reviewer's previous comment, our focus on the winter and summer seasons was intentional, as we observed that the most intense Red Sea MHWs occurred predominantly during these seasons.**

**In light of the Reviewer's concerns regarding the clarity of the selected periods, analysis, and spatial divisions, we have thoroughly revised the manuscript. We have taken into account the Reviewers' valuable suggestions to enhance clarity and ensure consistency throughout the document. We trust that these revisions address the concerns raised and provide a clearer understanding of our research.**

2010 event analysis it appears that is only for a winter event of the ten recorded during the year. Why?

**We selected this event because it was the most intense and longest winter MHW event of that year. This combination of factors makes it an interesting case for investigating its potential drivers (Lines 225-226 ).**

Please, do not use bold type fonts for the axis labels in the figures and don't use titles in plots if they can be explained in the caption.

**We have improved all the figures as suggested.**

Improve figure resolution for better readability. It's maybe because of the pdf conversion but carefully check all figures and use large enough fonts, especially if the text is placed inside plots.

**Thank you for pointing this out, we have improved all the figures, as suggested.**

Please, consistently use acronyms throughout the text. Take care especially when using MHW and "marine heat waves".

**Thank you, we have revised the manuscript to ensure acronyms are consistently used.**

Please, carefully review the text as there are some typos or misspellings.

**We have made our effort to correct all the typos in the revised manuscript.**

Line 172: an event can not be described by frequency, annual variability can.

**Yes, that's right, thank you for catching this mistake. It was corrected as: "MHWs can be described with a number of metrics, such as ....."**

Lines 172-173: Which is the difference between duration and total days. Maybe you are not referring to events but years?

**In this study, the duration refers to the period between the start and the end of a specific MHW event, while the MHW total days is the sum of all the MHW days over a period of time for example a year. In order to make this clearer and avoid any misunderstanding, we now provide a detailed definition for each MHW metric in the Methodology Section, as follows:**

**"MHWs can be described with a number of metrics, such as their duration (in day), which refers to the period between the start and end dates of a MHW event. Frequency (in events) indicates the number of MHW events that have occurred within a given year or period. Mean intensity (°C) is the average value of the temperature anomaly during the duration of a MHW event, while maximum intensity (°C) is the highest value of the temperature anomaly recorded during a MHW event. Cumulative intensity (°C.day) is the integrated temperature anomaly over the entire duration of a MHW event and is a measure of the overall intensity of the event. Total MHW days (MHWDs, in day) refers**

**to the total number of MHW days that have occurred in a given year/period (Hobday et al., 2016, 2018).”**

Lines 178-179: Is there any threshold for cold/warm years? Only the pos/neg sign of SSTA?

**The definition of "cold" and "warm" years is related to the SSTA variability and does not necessarily imply that the SSTA in those years was unusual or extreme. Specifically, warm (cold) years are identified as those that are warmer (colder) than the preceding or following years. In order to clarify this, we have added a paragraph in the Methodology Section about the cold/warm year definition, as follows:**

**"We further investigated the characteristics of MHWs during 'warm' or 'cold' periods. Specifically, we define warm periods as those that exhibit a pronounced positive SSTA compared to the long-term average, while cold periods are characterized by a pronounced negative SSTA. Warm years are identified as those that are warmer than the preceding and following year, and cold years as those that are colder than the year before and after. The definition of "cold" and "warm" years is relative to the SSTA variability and does not necessarily imply that the SSTA in those years was unusual or extreme."**

Line 180: How do you define a MHWD? A single day exceeding 90 percentile or a day belonging to a MHW event?

**The Total number of MHW days (MHWDs, in days) refers to the total number of MHW days that have occurred within a given year/period.**

Line 231: What does "non consistent" trend mean? Statistically? Spatially?

**We used the term "non consistent" to refer to events that were spatially different than those identified in other years. The sentence revised to enhance the clarity (line 277).**

Lines 246-249: Can't get the relevance of indicating the relatively cold/warm years in each period. It's just variability.

**You're correct that the definition of 'cold' and 'warm' years is relative to the sea SSTA variability and does not necessarily mean that the SSTA in those years was unusual or extreme. By examining the years that were warmer or colder than average, we aim to identify potential common spatial features in the MHW distribution.**

Lines 275-282: The description of atmospheric variables in the case of MHW events deserve a more extensive and detailed analysis. I suggest the authors to properly rewrite this part.

**We appreciate the Reviewer's suggestion. We have added a new paragraph in the Results and Discussion section to provide more details about the variability of atmospheric conditions before, during, and after a MHW event. We have also deepened our analysis by examining the temporal variability of SST in comparison with other atmospheric factors, including all heat flux components and relative humidity. A new figure has been also added to present these findings in the Supplementary Material (Fig. S8).**

**"To better understand how atmospheric forcings may have contributed to the development of this MHW event, the spatial averages of atmospheric variables before**

(February 3 to 7), during and after (March 10 to 15) the event were calculated and presented in Figure 14. Additionally, the time series of atmospheric variables averaged over the NRS (24° - 28° N and 34° - 39° E) during the event are presented in Supplementary Figure S8. Prior to the MHW event, the average SSTA in the NRS was about 1°C above average, while it was negative in the SRS and in the Strait of Bab El-Mandab. During the MHW event, the SSTA increased in the NRS and reached a local maximum of 4°C above the climatological average (Figure 14a-c). The spatial distribution of the average air temperature (Tair) showed higher values in the west (over Egypt, Eritrea and Ethiopia) than in the east (over Saudi Arabia) (Figure 14d-f). Over the NRS, the Tair increased by approximately 8°C compared to before the event. After the MHW, the Tair decreased but did not return to pre-MHW values (Figure 14d-f and Figure S8b). The mean sea level pressure (MSLP) maps showed an opposite distribution to Tair, with areas of high Tair having low MSLP and vice versa (Figure 14j-l). In addition, the average MSLP over the NRS decreased during the MHW event compared to before the event (Figure S8c). Before the MHW event, the winds blew from the eastern region and mainly flowed towards the SRS. During this event, the winds blew from the south and shifted to the west before reaching the NRS region, which experienced very low winds (Figures 14m-o and Figure S8d). Furthermore, the relative humidity rose by 10% over the NRS during the MHW period (Figure S8e).

In the RS, the latent heat flux (LHF) shares a similar spatial and temporal distribution with the net heat flux (Qt) (Nagy et al., 2021). The majority of the net surface heat exchange variability in the NRS is known to depend on the turbulent components of the surface flux, primarily the latent heat flux (Papadopoulos et al., 2013). In our case study, before the MHW event, the LHF ranged from -140 to -60 W/m², and the Qt ranged from -150 to -20 W/m², indicating that the ocean was losing heat to the atmosphere (Figure 14g-i and Figure S8f).

During the MHW, the combined effect of increased Tair, humidity and reduced winds led to a strong decrease in the ocean latent heat loss, signifying reduced heat loss to the atmosphere. Particularly during the days of the MHW onset and peak, the LHF fluctuated between -20 and -10 W/m². This decrease coincided with a slight increase in net solar radiation from 180 W/m² before the MHW to more than 200 W/m² during the MHW (Figure S8f). Accordingly, the heat exchange between the air and ocean reversed, causing a prolonged ocean heat gain, with Qt reaching up to 100 W/m², ultimately driving the MHW (Figure 14g-i).

In summary, our findings indicate that the late winter MHW event in the NRS was primarily driven by atmospheric forcing, specifically an increase in Tair and humidity, possibly linked to reduced winds. These atmospheric conditions collectively resulted in reduced LHF and a strong ocean heat gain, creating favourable conditions for MHW occurrence."

Line 356: lake?

Thank you for catching this typo, it is corrected to "lack".

---

## Author Comment (AC2)

Review of "Marine Heatwaves in the Red Sea and their Relationship to Different Climate Modes: A Case Study of the 2010 Events in the Northern Red Sea"

General Review

The authors have studied SST and MHWs in the Red Sea for the period 1982-2021. Mean MHW characteristics have been reported for the basin (and North/South sub-basins) and trends of SST anomaly and MHW frequency have been discussed over the study period. Potential links of SST anomaly and MHW frequency with certain climate indices have also been investigated. A more detailed analysis of selected sub-periods and groups of years has been shown, with a special focus on year 2010. The vertical extent and concurrent atmospheric conditions for a selected event that occurred within 2010 have also been investigated.

The topic of this work is important to address and I truly appreciate the efforts of the authors to provide a description of their results within a straightforward structure. There are however some parts that could be clarified and enriched to support some of the methodological choices made by the authors and substantiate their results:

A first concern is the criterion for detecting unusual behaviors of years and sub-periods. To my understanding, the definition of warm/cold years bears some issues, for which my concerns are detailed under specific comments (same concern for the "no-trend period"). In the same context, the choice for studying in detail the year 2010 has been slightly confusing while reading the text. It should be clear which specific findings justify the "exceptional character" of this year, thus motivating for a dedicated analysis.

**We greatly appreciate the Reviewer's thoughtful comments and constructive feedback. We have thoroughly considered all his/her suggestions and have addressed all concerns in the revised manuscript.**

**Regarding the definition of "cold" and "warm" years, it is relative to the SSTA variability and does not necessarily imply that the SSTA in those years was unusual or extreme. A paragraph was added in the Methodology Section to characterize the cold and warm years as follows:**

**"We further investigated the characteristics of MHWs during 'warm' or 'cold' periods. Specifically, we define warm periods as those that exhibit a pronounced positive SSTA compared to the long-term average, while cold periods are characterized by a pronounced negative SSTA. Warm years are identified as those that are warmer than the preceding and following year, and cold years as those that are colder than the year before and after. The definition of "cold" and "warm" years is relative to the SSTA variability and does not necessarily imply that the SSTA in those years was unusual or extreme."**

**Regarding the term "no-trend period", we agree with the Reviewer that its meaning was not clear. The entire paragraph has been revised to improve its clarity as follows:**

**"The analysis of SSTA between 1982 and 2021 revealed three distinct phases of variability in the Red Sea and its sub-basins (Fig. 5). The first phase, from 1982 to 1992, was characterized by negative SSTA on average. The second phase, between 1993 and 2015, showed a slow warming trend, but the SSTA fluctuated around zero, suggesting a relatively stable period with increased inter-annual variability. The third phase, from 2016 to 2021, was marked by a rapid increase in SSTA, with the anomaly consistently remaining positive."**

**Regarding the justification for studying this particular event, it was the most intense event of that year, longest winter event and occurred during the winter season, making it**

an interesting case to investigate its potential drivers. We have added new texts in the Introduction and Discussion Sections to provide further explanation for the selection of 2010 events as a case study.

Lines (119-121): "2010 was selected as a case study as it was one of the warmest years with highly frequent MHWs and has a different spatial distribution of SSTA and marine heatwave days (MHWDs) than the other warm years."

Lines (389-398): "The selection of 2010 as a case study for MHWs in the northern Red Sea is based on several reasons. Firstly, 2010 was one of the warmest years on record, with a high frequency of MHWs in the region. Secondly, the spatial distribution of SSTAs and MHWDs in 2010 was found to be different from that of other warm years. Thirdly, although the SRS is known to be warmer than the NRS through out the year (Fig. 2), in 2010 the SSTA of the NRS was higher by more than 1°C than the SRS (Supplementary Figures S2. d). Therefore, this section aims to provide a detailed description of the spatial and vertical extent as well as the potential atmospheric drivers of the intense MHW event that occurred in the NRS in 2010.

During both winter and summer of 2010, the NRS experienced ten MHW events (Fig. 12). These included one severe event in February and March (Category III), one strong event between October and December (Category II), and several moderate events (Category I). In this section, we will provide a detailed analysis of the most intense and longest winter MHW event that occurred in the NRS."

In addition, the choice of MHW frequency (being a tricky MHW property especially for basins with high warming rates) should probably be better justified, as the authors use only this MHW parameter to investigate links between MHWs and climate indices.

**We appreciate the Reviewer's comment. We found that both MHW frequency and days show the same spatial correlation patterns with the examined climate modes, albeit with slightly different values of the correlation coefficients. To avoid redundancy, we decided to present only the results for MHW frequency in the manuscript as it had the higher correlations with the climate modes. In addition, we also calculated the correlation between SSTA and climate modes to obtain a more comprehensive understanding of their relationship.**

**We agree with the Reviewer that this explanation can be useful in the text, which is now included in the Methodology Section as follows:**

"To gain a deeper understanding of the relationship between the different climate modes and the occurrence of MHWs over the last four decades in the RS, spatial correlations were examined. The climate modes considered in this study are the Oceanic Niño Index (ONI), the East Atlantic/West Russia Pattern (EATL/WRUS), the Atlantic Multidecadal Oscillation (AMO), the North Atlantic Oscillation (NAO) and the Indian Ocean Dipole (IOD). The correlation maps were calculated using the Pearson correlation coefficient (r), a widely used method for measuring linear correlations between two variables (Kirch, 2008; Patten and Newhart, 2017). The Pearson correlation coefficient ranges from -1 to 1, where -1 stands for a perfect negative correlation, 1 indicates a perfect positive correlation and 0 for no correlation. In this study, we calculated the correlation maps

between the annual time series of each climate mode and the annual MHWs/SSTA in the RS. The MHWs were identified and characterized using a set of metrics, such as their duration, frequency, mean intensity, maximum intensity, cumulative intensity and total days (as described in the previous Section). We calculated the correlation between the different annual climate modes and annual MHW metrics, in particular frequency, duration and total days. The results showed a consistent spatial correlation pattern with MHW different metrics, while the correlation coefficients varied only slightly. For the sake of brevity, we present only the correlation results with MHW frequency and SSTA in our results. The MHW frequency was chosen for presentation due to its slightly higher correlation compared to MHW duration and total days. To test the significance of the correlations, a two-tailed t-test was used (Patten and Newhart, 2017). The t-test is a statistical hypothesis test that compares the means of two samples and determines whether they differ significantly from each other. Finally, we also compared the time series between different climate modes and the frequency of MHWs in the RS and its sub-basins. By analyzing the correlation maps and the significance of the correlations, we can gain insights into the potential co-variability between MHWs in the RS and larger-scale climate variability."

A major concern is the interpretation of the atmospheric conditions and surface heat exchanges during the examined MHW. Causal links are reported in several parts within the text without sufficient analysis to support them (again, my concerns are detailed in the specific comments provided below).

**The authors appreciate the Reviewer's insightful comment, which has significantly improved the Discussion Section of our manuscript. We have addressed each specific comment provided below in detail.**

Finally, some language issues would greatly improve the manuscript if corrected. Some have been detected and reported under specific lines but further checks should be made (e.g. syntax issues or typos in text and captions). Red Sea or RS (same for NRS, SRS) should be consistently used throughout the text. I also strongly suggest a more conservative approach in wording that suggests a cause-and-effect relationship where results are not sufficient to support such conclusions.

**Thank you for pointing this out, we have revised and modified the manuscript as suggested and made our best to improve language and correct any typos in the revised text.**

On these grounds, I would like to kindly encourage the authors to send a revised version of their work and perceive this review as constructive feedback towards improving their already interesting work and important contribution to the MHW literature.

**We once again would like to thank the Reviewer for his/her kind comments on our manuscript and very valuable evaluation. We have taken all suggested comments into consideration and revised the Manuscript accordingly.**

Specific comments:

Abstract

Line 15: Why frequency?

**We found that both MHW frequency, duration and total days show the same spatial correlation patterns with the examined climate modes, albeit with slightly different values of the correlation coefficients. To avoid redundancy, we decided to present only the results for MHW frequency in the manuscript as it had the higher correlations with the climate modes. In addition, we also calculated the correlation between SSTA and climate modes to obtain a more comprehensive understanding of their relationship.**

**We agree with the Reviewer that this explanation can be useful in the text, which is now included in the Methodology Section as follows:**

**"To gain a deeper understanding of the relationship between the different climate modes and the occurrence of MHWs over the last four decades in the RS, spatial correlations were examined. The climate modes considered in this study are the Oceanic Niño Index (ONI), the East Atlantic/West Russia Pattern (EATL/WRUS), the Atlantic Multidecadal Oscillation (AMO), the North Atlantic Oscillation (NAO) and the Indian Ocean Dipole (IOD). The correlation maps were calculated using the Pearson correlation coefficient (r), a widely used method for measuring linear correlations between two variables (Kirch, 2008; Patten and Newhart, 2017). The Pearson correlation coefficient ranges from -1 to 1, where -1 stands for a perfect negative correlation, 1 indicates a perfect positive correlation and 0 for no correlation. In this study, we calculated the correlation maps between the annual time series of each climate mode and the annual MHWs/SSTA in the RS. The MHWs were identified and characterized using a set of metrics, such as their duration, frequency, mean intensity, maximum intensity, cumulative intensity and total days (as described in the previous Section). We calculated the correlation between the different annual climate modes and annual MHW metrics, in particular frequency, duration and total days. The results showed a consistent spatial correlation pattern with MHW different metrics, while the correlation coefficients varied only slightly. For the sake of brevity, we present only the correlation results with MHW frequency and SSTA in our results. The MHW frequency was chosen for presentation due to its slightly higher correlation compared to MHW duration and total days. To test the significance of the correlations, a two-tailed t-test was used (Patten and Newhart, 2017). The t-test is a statistical hypothesis test that compares the means of two samples and determines whether they differ significantly from each other. Finally, we also compared the time series between different climate modes and the frequency of MHWs in the RS and its sub-basins. By analyzing the correlation maps and the significance of the correlations, we can gain insights into the potential co-variability between MHWs in the RS and larger-scale climate variability."**

Line 16: Why 2010?

**The selection of 2010 as a case study for MHWs in the northern Red Sea is based on several reasons. Firstly, 2010 was one of the warmest years on record, with a high frequency of MHWs in the region. Secondly, the spatial distribution of SSTAs and MHWDs in 2010 was found to be different from that of other warm years. Thirdly,**

although the SRS is known to be warmer than the NRS through out the year, in 2010 the SSTA of the NRS was higher by more than 1°C than the SRS.

Line 17: "*...warming trend was observed that began in 1994 and has intensified significantly since 2016...*" Looking at the time-series of Fig. 5 I cannot understand how this conclusion is obtained. This also contradicts what the authors call a "no-trend-period" beginning at 1992

**We agree with the Reviewer that the warming trend (0.45 ± 0.2 °C/decade) is difficult to identify in Fig. 5. The reader can better identify this from Figure S2&4 in the Supplementary Figures. We have added this reference in the revised text to support our claim and make it clearer for the reader.**

**Regarding the term "no-trend period", we also agree with the Reviewer that its meaning was not clear in the initial text. The entire paragraph has been revised to improve its clarity as follows:**

**"The analysis of SSTA between 1982 and 2021 revealed three distinct phases of variability in the Red Sea and its sub-basins (Fig. 5). The first phase, from 1982 to 1992, was characterized by negative SSTA on average. The second phase, between 1993 and 2015, showed a slow warming trend, but the SSTA fluctuated around zero, suggesting a relatively stable period with increased inter-annual variability. The third phase, from 2016 to 2021, was marked by a rapid increase in SSTA, with the anomaly consistently remaining positive."**

Line 19: "*heat days*" here mean MHW days? Not clear

**Thank you for pointing this out. The correct term is "MHW days". We have corrected this typo and further revised the entire manuscript to ensure consistency in the use of abbreviations.**

Line 23: "*while it was highest*" should better be rephrased

**Thank you for the comment. The whole sentence has been revised as follows:**

**"The annual MHW frequency in the NRS peaked in 2010, 2018, 2019 and 2021, while in the SRS the highest frequency occurred in 1998 and from 2017 to 2021."**

Line 23-24: could probably be removed from abstract to focus on results that come right next

**The sentence has been removed as suggested.**

Line 24: Why frequency? What about other properties?

**We have chosen to focus on MHW frequency for the interannual variability analysis in order to maintain consistency with the other parts of the study. Additionally, we found that all MHW metrics (frequency, duration, and total days) showed a similar accelerated trend in the last decade, so the same conclusion would have been reached regardless of the chosen metric. Following the Reviewer's comment, we have added an additional figure in the Supplementary Material to show the trends in the MHW duration and total days (Fig. S1) and the corresponding discussion revised as follow:**

**Lines (284-290): "Furthermore, the trends in MHW duration and total days also displayed spatial variation in the RS (Supplementary Figure S1). The MHW duration spatial trend fluctuated between 2 to more than 10 days per decade, with an average temporal trend of 2.8 ± 1.25 days/decade (Fig. S1 a-b). The highest MHW duration trends were observed in the central RS, some parts of the NRS, and the northern part of the Suez and Aqaba Gulfs. The MHW total days trend ranged between 15 to more than 30 days per decade, with an average temporal trend of 20.04 ± 6.88 days/decade (Fig. S1 c-d). The highest trends were observed in the SRS, some parts of the NRS, and the northern part of the Suez and Aqaba Gulfs. Notably, for all the MHW metrics (frequency, duration, and total days), there is an accelerated positive trend that is more pronounced in the last decade."**

Line 27: same as before

**We found that both MHW frequency, duration and total days show the same spatial correlation patterns with the examined climate modes, albeit with slightly different values of the correlation coefficients. To avoid redundancy, we decided to present only the results for MHW frequency in the manuscript as it had the higher correlations with the climate modes. In addition, we also calculated the correlation between SSTA and climate modes to obtain a more comprehensive understanding of their relationship.**

Introduction

Line 48: A smoother transition from the 1st to the 2nd paragraph would improve readability

**Thank you for pointing this out. We have revised the text accordingly as follows:**

**The increasing risk of MHWs on ecosystems and economies requires a thorough understanding of their causes, especially in vulnerable areas such as the Red Sea, which harbors a unique ecosystem and is of great political and economic importance.**

Line 59-61: I would suggest: "However, only a few studies have investigated MHWs in the Red Sea (Genevier et al., 2019; Bawadekji et al., 2021; Mohamed et al., 2021) and up to date there have been no studies investigating the link between climate patterns and the occurrence of MHWs in the Red Sea region"

**The sentence was revised as suggested.**

Line 67: Maybe better present separately any literature on future projections (now reported among results for past trends)

**We appreciate the Reviewer's suggestion; we have moved the sentence to the end of the paragraph to separate literature references from the future projections.**

Line 77: marine heatwaves -> MHWs

**Thank you. This has been corrected here and throughout the manuscript.**

Line 90-91: typos: un -> in, The -> the. Also use articles before each index consistently, or simply list indices, e.g., […] and their correlation with the following climate indices: AMO, IOD,…etc

**Thank you, all the typos were corrected.**

Line 90-92: Maybe move this sentence after introducing all different indices

**Thank you for the suggestion. The sentence was moved as suggested.**

Line 119: why frequency?

**We found that both MHW frequency, duration and total days show the same spatial correlation patterns with the examined climate modes, albeit with slightly different values of the correlation coefficients. To avoid redundancy, we decided to present only the results for MHW frequency in the manuscript as it had the higher correlations with the climate modes. In addition, we also calculated the correlation between SSTA and climate modes to obtain a more comprehensive understanding of their relationship.**

Line 121: As in general comment, choose RS or Red Sea

**Thank you. We have checked the entire manuscript to ensure that abbreviations are consistently used.**

Line 93-4: A reference should better be added here

**Thank you. A reference has been added as suggested.**

2.1 Datasets

Line 116-118: please, rephrase

**We have rephrased the study aim as follows:**

**"The objective of this study is to conduct a comprehensive analysis of the spatial and temporal variability of MHWs in the Red Sea (RS) and to identify its regional patterns. The study also aims to investigate the potential links between various climate modes, particularly the AMO, the IOD, the EATL/WRUS pattern, the NAO and the ONI, with the annual sea surface temperature anomaly (SSTA) and the annual frequency of MHWs in the RS. As a case study, we will focus on the 2010 MHWs and provide a detailed description of the spatial and vertical extent and potential atmospheric drivers of the intense event in that year. The motivation for selecting 2010 as a case study is that it was one of the warmest years with highly frequent MHWs and has a different spatial distribution of SSTA and marine heatwave days (MHWDs) than the other warm years.**

**To accomplish these goals, the study is structured into four main sections. The first section focuses on the characteristics and trends of SSTs and MHWs in the RS between 1982 and 2021. The second section examines the interannual variability of SSTA and MHWs over the last four decades in the RS and its northern and southern basins. The third section explores the relationship between SSTA/MHWs in the RS and the different**

**climate modes. Finally, in the fourth section, a case study of the 2010 MHW events in the NRS is presented."**

Line 118-119: "*The study also aims to investigate the correlations between the different climate modes with the annual sea surface temperature anomaly (SSTA) and the annual frequency of MHWs in the Red Sea* " Why this parameter? Using only MHW frequency should better be justified.

**We found that both MHW frequency, duration and total days show the same spatial correlation patterns with the examined climate modes, albeit with slightly different values of the correlation coefficients. To avoid redundancy, we decided to present only the results for MHW frequency in the manuscript as it had the higher correlations with the climate modes. In addition, we also calculated the correlation between SSTA and climate modes to obtain a more comprehensive understanding of their relationship.**

**However, we agree with the Reviewer that this was not clear in our initial manuscript. We have included one paragraph to make this clearer for the readers in our revised text as discussed in the previous responses.**

2.2 Methods of analysis:

Line 165: please, rephrase sentence

**We have rephrased the sentence as follows:**

**"Marine heatwaves (MHWs) can be characterized by different methods, each of which has its advantages and disadvantages. These methods include the use of fixed thresholds, relative thresholds, and seasonally varying thresholds (Hobday et al., 2016; Mohamed et al., 2022)."**

Line 179: Define strong positive/negative, it is not clear

**Thank you for the suggestion, the whole paragraph has been revised as follows:**

**"We further investigated the characteristics of MHWs during 'warm' or 'cold' periods. Specifically, we define warm periods as those that exhibit a pronounced positive SSTA compared to the long-term average, while cold periods are characterized by a pronounced negative SSTA. Warm years are identified as those that are warmer than the preceding and following year, and cold years as those that are colder than the year before and after. The definition of "cold" and "warm" years is relative to the SSTA variability and does not necessarily imply that the SSTA in those years was unusual or extreme."**

Line 190: What would be *unusual* for MHWs and climate modes? The motivation is not very clear here, and maybe it should be introduced earlier in the manuscript, justifying why year 2010 is a case study.

**Thank you for the suggestion. As we discussed in our previous responses, we have added a relevant discussion in the Introduction and Discussion Sections to justify our selection of 2010 as a case study (Lines 119 – 121 and 389 – 398 in the revised MS).**

Line 191-2: Here you focus on events falling entirely within 2010? Or also partly e.g., the ones peaking within the year?

**In this study we calculated and categorized all the events occurred in 2010, but we described in detail only the late winter event which was the most intense and longest winter event in that year.**

Line 191: Therefore probably does not fit here

**The sentence has been removed.**

Line 194: You mean when SST exceeds the threshold by #X (SSTthres-SSTclim), instead of #X (SSTthres), right? Also, simply exceeding the threshold shows that MHW occurs (of any category, not a moderate one). So please revise the sentence accordingly.

**We thank the Reviewer for pointing this out. The categories are based on the multiple of the local difference between the climatological mean and the climatological 90th percentile, which serves as a threshold for identifying MHW. Therefore, we corrected it in the text as follow:**

**"The MHW events were identified using the methodology described by Hobday et al., (2016) and then categorized into four intensity levels based on the multiple of the local difference between the climatological mean and the climatological 90th percentile, which serves as a threshold for identifying MHW. This anomaly varies by location and time of year. The magnitude of the scale descriptors was defined as follows: moderate (1–2×, category I), strong (2–3×, category II), severe (3–4×, category III) and extreme (>4×, category IV) (Hobday et al. 2018)."**

Line 194-5: Why these events?

**Because it was the most intense and longest winter event of that year, making it an interesting case to investigate its potential drivers.**

3. Results

Line 209: […] in the RS.

**Thank you. It is added.**

Line 218: better write "with longer MHW durations"

**Thank you. It is modified as suggested.**

Line 220: Here you mean the most intense?

**Corrected. The text now reads: "The most intense MHWs were observed in the NRS…."**

Line 220-22: "*Both the mean…spatial variability (Fig. 3e, f)*" I cannot see a "similar pattern". There is a north-south gradient in both fields but there are also non negligible differences (Gulfs of Suez/Aqaba, coastal areas in central and southern RS, Strait of Bab El-Mandeb).

**Thank you for the suggestion, the sentence is rewritten as follows:**

==**"Furthermore, the mean cumulative MHW intensity (Icum) and the total number of heat days exhibited a north-south gradient, with higher values in the northern Red Sea region (Fig. 3e, f).**==**"**

Line 222: Figure 3e

**Thank you it is corrected.**

Line 231: What does non consistent mean here? Please, revise for clarity.

**Thank you for pointing this out. Here "non consistent" means spatially non consistent. for more clarity it is replaced by "not uniform" to avoid confusion.**

Line 232: Specify the area geographically (in relation or not to its bathymetry, as you prefer) but rephrase, as the deep waters cannot exhibit SST trends :)

**Thank you for the suggestion, the whole paragraph was revised as follows:**

==**"The trends of SST and MHW frequency in the RS are not uniform and range between 0.1 to 0.5°C/decade for SST and 0.5 to 2 events/decade for MHW frequency as shown in Figure 2. The NRS experienced high SST trends, with a maximum of about 0.4°C/decade between 25°N and 28°N. However, the highest SST trends were observed between 16°N and 25°N, with a gradient that increased towards offshore waters and exceeded 0.45°C/decade. In contrast, the southernmost part of the RS, in the vicinity of Bab El-Mandab Strait, and the Gulfs of Suez and Aqaba exhibited the lowest SST trends, with trends below 0.15°C/decade.**==**"**

Line 235-37: This statement is probably too far-fetched. In addition, there areas ~17degN with high frequency trends in both figs. Most importantly, why is frequency the appropriate matric to derive such a conclusion? Why not compare trends of other parameters against SST trends?

**We thank the Reviewer for the comment. The sentence is indeed confusing. We therefore decided to removed it and focus on the other conclusions.**

3.2

Line 240: Are trends stat. significant?

**Yes, all the trends presented in this work are estimated using the least squares method (Wilks, 2019) and their statistical significance is determined using the Modified Mann-Kendall test (MMK) at the 95% confidence level, which takes autocorrelation into account when assessing the significance of the trend (Hamed and Ramachandra Rao, 1998; Wang et al., 2020). This is now mentioned in the Methodology section in the revised Manuscript.**

Line 242-246: This separation into "*distinct phases of variability*" seems somewhat arbitrary. For instance, why the no-trend-period begins at 1992 and not 1996? Figure 5a shows from 1992 up to 1996 a clear warming trend (already present during the previous period).

**We thank the Reviewer for the comment. The "no-trend period" term is indeed confusing. This paragraph was rewritten as follows:**

==**"The analysis of SSTA between 1982 and 2021 revealed three distinct phases of variability in the Red Sea and its sub-basins (Fig. 5). The first phase, from 1982 to 1992, was characterized by negative SSTA on average. The second phase, between 1993 and 2015, showed a slow warming trend, but the SSTA fluctuated around zero, suggesting a relatively stable period with increased inter-annual variability. The third phase, from 2016 to 2021, was marked by a rapid increase in SSTA, with the anomaly consistently remaining positive."==**

Line 246: I have major concerns regarding the criterion used for detecting "unusually" cold (warm) years (i.e., colder (warmer) than the previous or following year) implies that SST variability is unusual.

**We thank the reviewer for the comment. We now mention in the Methodology Section that the definition of "cold" and "warm" years is relative to the SSTA variability and does not necessarily imply that the SSTA in those years was unusual or extreme (Lines 197-201).**

Line 250-255: Try to follow clearly explained criteria for characterizing temporal periods instead of phrases "generally warm", "generally high", "particularly warm" mentioned in this paragraph. This comment applies for the entire manuscript (when characterizing years, sub-periods and trends).

**Thank you for the comment. The paragraph was rewritten as follows:**

==**"The analysis of SSTA between 1982 and 2021 revealed three distinct phases of variability in the Red Sea and its sub-basins (Fig. 5). The first phase, from 1982 to 1992, was characterized by negative SSTA on average. The second phase, between 1993 and 2015, showed a slow warming trend, but the SSTA fluctuated around zero, suggesting a relatively stable period with increased inter-annual variability. The third phase, from 2016 to 2021, was marked by a rapid increase in SSTA, with the anomaly consistently remaining positive. Moreover, the monthly SSTA data for the RS and its sub-basins show a clear warming trend that began in 1994, with an initial SSTA of approximately $0.5°C$. The SSTA remained relatively stable for a few years, but then increased rapidly after 2016, reaching an SSTA of $1.5°C$ or higher (Supplementary Figure S2.a-c). This finding is consistent with the results of Raitsos et al. (2011). It was also observed that during years when the RS experienced cold SSTs, the NRS was warmer than the SRS, especially during the winter and autumn months. Conversely, during years when the RS experienced warm SSTs, the SRS was warmer than the NRS. The year 2010 was one of the warmest years in the RS, but it was particularly warm in the NRS, with an SSTA difference between the NRS and SRS of over $1°C$ (Supplementary Figures S2. d). Further analysis of the SSTA trends revealed a 10-year alternation cycle (Supplementary Figures S3-4). Between 1982 and 1991, the highest trends were observed in the NRS, with an average trend of $0.56°C/Decade$, while the average trend in the SRS was $0.26°C/Decade$. From 1992 to==**

**2001, the spatial pattern of the SSTA trend was altered, with the highest trends observed in the SRS, with an average trend of 0.57°C/Decade, and lower trends in the NRS, with an average trend of 0.30°C/Decade. From 2002 to 2011, the highest trends were again observed in the NRS, with an average trend of 0.45°C/Decade, while the SRS experienced no trend in the SSTA during this period. Finally, over the last decade of the study period (2012-2021), the SRS had higher trends in the SSTA than the NRS, with an average trend of 1.35°C/Decade for the SRS and 0.89°C/Decade for the NRS."**

**In order to enhance the clarity of characterizing years, sub-periods, and trends, we have thoroughly revised the manuscript. We have taken into account the Reviewers' valuable suggestions to ensure clarity and consistency throughout the document. We trust that these revisions address the concerns raised and provide a clearer understanding of our research.**

Line 258-9: "*In warm years, the SRS and the northern regions of the Gulfs of Suez and Aqaba had the highest SSTA and also the highest number of MHWDs (Fig. 7) ".* This is not true, as Gulfs of Suez and Aqaba show minimum SSTA during the "warm" years, which is not the case for MHWDs in these regions.

**The Reviewer is right. The paragraph was revised as follows:**

**"During both the warm and cold years, the spatial distribution of the average SSTA and MHWDs was analyzed, as shown in Figures 6-8. In the cold years, the NRS and the Strait of Bab El-Mandab had the highest SSTA and MHWDs (Figure 6). However, in the warm years, the SRS had the highest SSTA, and the SRS and the northern regions of the Gulfs of Suez and Aqaba had the highest number of MHWDs (Figure 7). The year 2010 was an exception among the warm years, with a distinct spatial distribution of SSTA and MHWDs. In 2010, the NRS and the Gulfs of Suez and Aqaba had the highest SSTA and MHWDs (Figure 8)."**

Line 260: "*2010 was a unique year...*" How do we know this? To my understanding, comparing 2010 vs a warm-years average cannot support that 2010 shows an unusual behavior. I guess that other individual years may show similar spatial SSTA/MHWD patterns with 2010, or be very different from the warm-year average.

**Thank you for this comment. We have checked the spatial distribution of the SSTA and MHWDs for each warm and cold year separately before calculating the averages. Our results indicate that all the warm years, except for 2010 and 2018, have the same spatial distribution for the SSTA and MHWDs, with higher values observed in the SRS. In 2010, the SSTA and MHWs were only present in the NRS, while in 2018, the values were recorded in both the NRS and SRS, but the values in the southern basin were higher than the ones in the north, which still follows the spatial pattern observed in the other warm years.**

Line 263: I would rephrase: The inter-annual variations of MHWs frequency in the RS are shown in Fig. 9.

**Thank you. It is modified as suggested.**

Line 272-4: "*indicate that the rapid increase in SST in the RS has caused…*" Maybe comment here (and/or later in discussion) on how the your methodological choice for using a fixed climatological baseline affects the event detection (being dependent on both the warming trend of the RS and internal variability). Also: "*…are expected to increase…*" I would avoid this statement.

**Thank you for the comment. Regrading the use of this baseline, we added some Discussion and Conclusions explain it as follows:**

**Lines (250 - 260): "The marine environment is influenced by both natural variability and global warming trends. Over time, the difference between a fixed baseline and current temperatures can widen, leading to an increasing number of detected MHWs. This temporal shift can complicate long-term studies of MHW trends and their impacts on marine ecosystems. However, using a fixed baseline simplifies the methodology for detecting MHWs and avoids the complexity that could arise from periodically updating the baseline, which could introduce variability and reduce the clarity of the detection process. A fixed baseline provides a standardized reference period that ensures consistency of MHW detection and analysis across different studies and time periods (e.g. (Genevier et al., 2019; Cheng et al., 2023)). This consistency allows for straightforward comparison between MHW events detected using the same criteria. In addition, a fixed baseline serves as a historical benchmark against which current and future MHW events can be measured. This allows an assessment of how current conditions deviate from a known historical norm. To account for SST variability during the study period and to emphasize the impact of external forcing on marine ecosystems, we calculated MHW characteristics in the RS based on 40 years of climatology (1982-2021)."**

**Lines (455 - 459): "Considering the results of this study and the observed trends of MHWs in the region, it is recommended that future work considers an analysis of MHW trends based on different baselines. This comparison is particularly important when it comes to projecting future MHWs under different global warming scenarios, as the selection of an appropriate baseline is of utmost importance for the detection of future MHWs and the calculation of their trends."**

**Regarding the second part of the Reviewer's question, a paragraph is modified as follows:**

**"The findings of this study suggest that the recent rapid increase in SST in the RS has led to a positive trend in the occurrence of MHWs in the region. These findings are consistent with those of previous studies, such as Bawadekji et al., (2021) and Mohamed et al., (2021), which have also documented the increasing trend of MHWs in the RS. This trend is expected to continue in the future, as global warming is projected to cause further increases in SSTs, both in the RS and in other regions around the world."**

Line 279: "*This high heat flux has caused...*" Fig. S3 cannot support a causal link between Tair and heat flux.

Line 280: "*Since the wind in these years was not sufficient to support the cooling of the SST (S4), this excess heat absorbed by the ocean from the atmosphere likely led to the formation of several MHW events in the above mentioned years*" Is this an assumption or has it been examined also analyzing the heat flux components? Weak winds are potentially associated with reduced latent heat loss from the sea surface leading to increased net heat flux. Such an analysis

should be presented if a conclusion is to be made on the role of low winds in causing/maintaining MHW conditions during specific periods. Importantly, S4 does not show a significant decrease in wind speed in the examined years/regions (2010 in the NRS and 1998 in the SRS).

**In response to the Reviewer's previous 2 comments, we have rewritten the entire paragraph to enhance clarity and avoid confusion. The revised paragraph is as follows:**

"To gain a better understanding of the atmospheric conditions associated with the years with the highest frequency of MHWs in the RS and its subregions, we compared the atmospheric variables with the annual MHW frequency. The annual anomalies of total heat flux (Qt), air temperature and wind speed are shown in Supplementary Figures S5-S7. Our analysis revealed that the years with a high MHW frequency were characterized by a specific set of atmospheric conditions. In particular, these years were characterized by reduced wind speeds and high air temperatures and anomalously high Qt which may have contributed to the frequent occurrence of MHWs in the RS during these years ."

3.3

Line 290: Not true for MHW frequency (Fig. 10b), please check

**Thank you for the comment, the sentence is modified as follows:**

"The AMO index showed a highly significant positive correlation with both SSTA and MHW frequency across the whole RS, with a correlation coefficient of greater than 0.7 for SSTA and greater than 0.5 for MHW frequency (Figures 10a and b)."

Line 294: "between 0.2-0.4" is for SSTA or freq?

**This correlation is for the MHW frequency. For more clarity, the sentence is modified as follows:**

"The IOD index also showed a positive correlation with both SSTA and MHW frequency in RS, with a correlation coefficient of greater than 0.5 for SSTA and ranging between 0.2 and 0.4 for MHW frequency (Figures 10c and d)."

Line 295-6: "*...in the deepest part of the RS..*" Not sure I understand the need to link with deep water formation areas in this sentence.

**Thank you for the comment. The sentence is modified as follows:**

"This correlation was strongest in the NRS and in the offshore areas of the central and southern RS."

Line 297: were -> was

**Thank you for the comment. It is revised as suggested.**

Line 298: in -> for , The later -> The latter

**Thank you for the comment. It is revised as suggested.**

Line 305-6: "*These are the same areas…*" This is true for the western coast around 14-18degN, but areas do not coincide

**Thank you for the comment. It is revised as follows:**

==**"The correlation between the ONI and MHW frequency was not significant, except for a correlation of about -0.2 on the western coast of the SRS (Figure 10j)."**==

Line 307-318 I suggest avoiding the term coincidence in this discussion. I have the impression that in some cases the actual relationship among the examined parameters (as shown from results) is not as strong as implied in certain parts of the text e.g. line 315.

**Thank you for the comment. It is revised as suggested.**

3.4

Line 329-330 Please, rephrase for clarity

**Thank you for the comment. It is revised as follows:**

==**"To better understand the dynamics of the event, the anomaly of water column temperature and the occurrence of the MHW were separately calculated at different depth levels (surface, 25 m, 55 m, 110 m and 130 m) and further compared to the mixed layer depth (MLD), as shown in Figure 13."**==

Line 337: "*especially in the surface layers*" Where do we see this?

**We agree with the Reviewer that the sentence was misleading, it is changed as follows:**

==**"In addition, the results revealed a strong negative relationship between upper layer temperature and MLD, with a thin mixed layer aligned with the days of the higher water temperature."**==

Line 341: But at 110m the event did not take place at the peak of the surface event (as I see in Fig. 14).

**The MHW event occurred in the surface layer between February 12 and March 19 and peaked on March 12, 2010. At 110 m depth, the duration of the MHW was shortest compared to the upper layers and took place from February 26 to March 17, which is around the peak day of the surface MHW.**

Line 343: "*indicating that…*" I would remove this part as this is not clear from the data and is only based on a single event

**Thank you for the comment. The sentence is removed.**

Line 345: maybe say reaching locally 4deg?

**Thank you for the comment. It is revised as suggested.**

Line 346: western coasts (not western Africa)

**Thank you for the comment. It is revised as suggested.**

Line 347: "*This increase in Tair led to…*" Such a cause-effect relationship is not supported. See also my previous concern on Line 280.

Line 355-6: Same concern as above. The interpretation of the increased net heat flux should be based on the bahavior of its components. The increase in Tair and total heat flux during the event cannot support that the event is "due to large amount of heat absorbed by the ocean". There might be significantly suppressed heat loss (probably caused by the weakened winds) being responsible for the observed heat balance during the event. Reduced latent heat loss could also be associated with drier than usual air masses but all this should be investigated in order to support a causal link between the anomalously warm SST, heat flux and other physical parameters (e.g., wind).

**We appreciate the Reviewer's suggestion. Replying to the Reviewer previous 2 comments, we have added a new paragraph in the Results and Discussion section with more details about the variability of atmospheric conditions before, during, and after the MHW event. We have also deepened our analysis by examining the temporal variability of SST in comparison with other atmospheric factors, including all heat flux components and relative humidity. A new figure has been also added to present these findings in the Supplementary Material (Fig. S8).**

**"To better understand how atmospheric forcings may have contributed to the development of this MHW event, the spatial averages of atmospheric variables before (February 3 to 7), during and after (March 10 to 15) the event were calculated and presented in Figure 14. Additionally, the time series of atmospheric variables averaged over the NRS (24° - 28° N and 34° - 39° E) during the event are presented in Supplementary Figure S8. Prior to the MHW event, the average SSTA in the NRS was about 1°C above average, while it was negative in the SRS and in the Strait of Bab El-Mandab. During the MHW event, the SSTA increased in the NRS and reached a local maximum of 4°C above the climatological average (Figure 14a-c). The spatial distribution of the average air temperature (Tair) showed higher values in the west (over Egypt, Eritrea and Ethiopia) than in the east (over Saudi Arabia) (Figure 14d-f). Over the NRS, the Tair increased by approximately 8°C compared to before the event. After the MHW, the Tair decreased but did not return to pre-MHW values (Figure 14d-f and Figure S8b). The mean sea level pressure (MSLP) maps showed an opposite distribution to Tair, with areas of high Tair having low MSLP and vice versa (Figure 14j-l). In addition, the average MSLP over the NRS decreased during the MHW event compared to before the event (Figure S8c). Before the MHW event, the winds blew from the eastern region and mainly flowed towards the SRS. During this event, the winds blew from the south and shifted to the west before reaching the NRS region, which experienced very low winds (Figures 14m-o and Figure S8d). Furthermore, the relative humidity rose by 10% over the NRS during the MHW period (Figure S8e).**

**In the RS, the latent heat flux (LHF) shares a similar spatial and temporal distribution with the net heat flux (Qt) (Nagy et al., 2021). The majority of the net surface heat exchange variability in the NRS is known to depend on the turbulent components of the surface flux, primarily the latent heat flux (Papadopoulos et al., 2013). In our case study,**

**before the MHW event, the LHF ranged from -140 to -60 W/m², and the Qt ranged from -150 to -20 W/m², indicating that the ocean was losing heat to the atmosphere (Figure 14g-i and Figure S8f).**

**During the MHW, the combined effect of increased Tair, humidity and reduced winds led to a strong decrease in the ocean latent heat loss, signifying reduced heat loss to the atmosphere. Particularly during the days of the MHW onset and peak, the LHF fluctuated between -20 and -10 W/m². This decrease coincided with a slight increase in net solar radiation from 180 W/m² before the MHW to more than 200 W/m² during the MHW (Figure S8f). Accordingly, the heat exchange between the air and ocean reversed, causing a prolonged ocean heat gain, with Qt reaching up to 100 W/m², ultimately driving the MHW (Figure 14g-i).**

**In summary, our findings indicate that the late winter MHW event in the NRS was primarily driven by atmospheric forcing, specifically an increase in Tair and humidity, possibly linked to reduced winds. These atmospheric conditions collectively resulted in reduced LHF and a strong ocean heat gain, creating favourable conditions for MHW occurrence."**

Line 356: lake you mean lack?

**Thank you for catching this typo. It is now corrected to "lack".**

4. Conclusions

Line 368: occur -> are

**Thank you for the comment. It is revised as suggested.**

Line 370-2: See previous comments (eg  Line 355-6)

**Thank you for the comment. It is revised as suggested and modified as mentioned in the previous comments.**

Line 372: See previous comment (eg Line 260) on the exceptional character of 2010.

**Thank you for the comment. It is revised as suggested and modified as mentioned in the previous comments.**

Line 374: "*2010 breaks the trend with the highest values ever recorded in the NRS*" This should be corrected as Fig. 5 shows that 2021 presents higher SSTA than 2010. It should be clear in terms of which characteristics the year 2010 shows an "exceptional" character (and on these grounds explain the motivation for this case study).

**Thank you for the comment. It is revised as suggested and modified as mentioned in the previous comments.**

Line 380: "*resulting in…*" Please, see previous concerns on assuming causal links

**Thank you for the comment. It is revised as suggested and modified as mentioned in the previous comments.**

Fig. 12 Is this plot based on mean basin (NRS) SST? Also, see previous comment on explaining the different categories based on threshold exceedance)

**Yes, this plot is calculated from the mean NRS sea surface temperature. The Figure title is revised as suggested.**

Fig. 14 Maps are for 2010 but this is not mentioned in caption. Also, the middle row is labeled as "event peak day" while caption says it corresponds to the average from 20 to 25 March. Please, revise accordingly

**Thank you for the comment. It is revised as suggested and modified as mentioned in the previous comments.**

Fig. 13 Title of panel (a): extend -> extent

**Thank you for catching this error. It is corrected as suggested.**

Supplement:

In S2, S3 and S4, captions state: "*The red shaded areas represent the years with the highest MHW frequency in each basin.*" Therefore, shouldn't exactly the same periods be shaded in these 3 figures?

**Thank you for the comment. Yes, it should be the same red regions for all the figures, the differences was due to error during plotting that is now corrected.**

---

## Author Response (AR2)

Author response:

Manuscript has greatly improved and concerns raised by the reviewers have been adequately answered. My recommendation is to accept with minor revision. Please fix the following question and revise the text accordingly.

I can not accept the definition of cold/warm years provided by the authors:

"We further investigated the characteristics of MHWs during 'warm' or 'cold' periods. Specifically, we define warm periods as those that exhibit a pronounced positive SSTA compared to the long-term average, while cold periods are characterized by a pronounced negative SSTA. Warm years are identified as those that are warmer than the preceding and following year, and cold years as those that are colder than the year before and after. The definition of "cold" and "warm" years is relative to the SSTA variability and does not necessarily imply that the SSTA in those years was unusual or extreme."

This definition could lead to an extreme year with 4ºC anomaly to be defined as cold if the preceding and following years are even more extreme with a 5ºC. This definition needs to be changed before publication. At least, do not refer to "cold" in this case. It is also confusing to define "warm periods" by the sign of the anomaly but a "cold year" could have positive SSTA.

**The Authors would like to thank the Reviewer for his/her kind comments on this paper and their very valuable evaluation.**

**We fully agree with the Reviewer that the definition could be confusing for the reader. Therefore, this part is modified in the Methodology and the Results Sections as follow:**

**Lines 194 – 200: "We further investigated the characteristics of MHWs during 'warm' or 'cold' years. Specifically, we define warm years as those that exhibit a pronounced positive SSTA compared to the long-term average, while cold years are characterized by a pronounced negative SSTA. The definition of "cold" and "warm" years is relative to the SSTA variability and does not necessarily imply that the SSTA in those years was unusual or extreme. Once the warm or cold years are identified, we calculate the average SSTA for those years by averaging the SSTA values over the entire years for each grid cell. Similarly, we calculate the MHWDs for the warm or cold years by averaging the MHWDs over those years for each grid cell. This gives us an indication of the overall spatial variability of the SSTA/MHWDs during the warm or cold years in our study period."**

**Lines 311 – 318: "Over the study period, there were years that were notably colder or warmer than the average for that period. The cold years were 1985, 1990, 1992, 1993, 1997, 2012 and 2013, while the warm years were 1991, 1995, 2010 and the last six years of the study period (2016-2021). During both the warm and cold years, the spatial distribution of the average SSTA and MHWDs was analyzed, as shown in Figures 6-8. In the cold years, the NRS and the Strait of Bab El-Mandab had the highest SSTA and MHWDs (Figure 6). However, in the warm years, the SRS had the highest SSTA, and the SRS and the northern regions of the Gulfs of Suez and Aqaba had the highest number of MHWDs (Figure 7). The year 2010 was an exception among the warm years, with a distinct spatial distribution of SSTA and MHWDs. In 2010, the NRS and the Gulfs of Suez and Aqaba had the highest SSTA and MHWDs (Figure 8)."**